# Beyond Match Maximization and Fairness: Retention-Optimized Two-Sided Matching

**Ren Kishimoto**[*]
Institute of Science Tokyo
Tokyo, Japan
kishimoto.r.ab@m.titech.ac.jp

**Rikiya Takehi**[*]
Waseda University
Tokyo, Japan
rikiya.takehi@fuji.waseda.jp

**Koichi Tanaka**
Keio University
Tokyo, Japan
kouichi_1207@keio.jp

**Masahiro Nomura**
Institute of Science Tokyo
Tokyo, Japan
nomura@comp.isct.ac.jp

**Riku Togashi**
CyberAgent
Tokyo, Japan
rtogashi@acm.org

**Yoji Tomita**
CyberAgent
Tokyo, Japan
tomita_yoji@cyberagent.co.jp

**Yuta Saito**
Hanuku-kaso, Co., Ltd.
Tokyo, Japan
saito@hanjuku-kaso.com

## Abstract

On two-sided matching platforms such as online dating and recruiting, recommendation algorithms often aim to maximize the total number of matches. However, this objective creates an imbalance, where some users receive far too many matches while many others receive very few and eventually abandon the platform. Retaining users is crucial for many platforms, such as those that depend heavily on subscriptions. Some may use fairness objectives to solve the problem of match maximization. However, fairness in itself is not the ultimate objective for many platforms, as users do not suddenly reward the platform simply because exposure is equalized. In practice, where user retention is often the ultimate goal, casually relying on fairness will leave the optimization of retention up to luck.

In this work, instead of maximizing matches or axiomatically defining fairness, we formally define the new problem setting of maximizing user retention in two-sided matching platforms. To this end, we introduce a dynamic learning-to-rank (LTR) algorithm called **M**atching for **Ret**ention (**MRet**). Unlike conventional algorithms for two-sided matching, our approach models user retention by learning personalized retention curves from each user's profile and interaction history. Based on these curves, MRet dynamically adapts recommendations by jointly considering the retention gains of both the user receiving recommendations and those who are being recommended, so that limited matching opportunities can be allocated where they most improve overall retention. Naturally but importantly, empirical evaluations on synthetic and real-world datasets from a major online dating platform show that MRet achieves higher user retention, since conventional methods optimize matches or fairness rather than retention.

## 1 Introduction

Predominantly, recommendations for two-sided matching platforms like online dating (Neve & Palomares, 2019; Pizzato et al., 2010b; Xia et al., 2015a) and recruitment (Jiang et al., 2020; Le et al., 2019; Yang et al., 2022), aim to maximize the overall number of matches between the two sides of users (Mine et al., 2013; Pizzato et al., 2010a; Palomares et al., 2021; Pizzato et al., 2010b; Qu et al., 2018). However, this objective leads to significant matching imbalances, where some users receive

---

[*]Equal contribution.

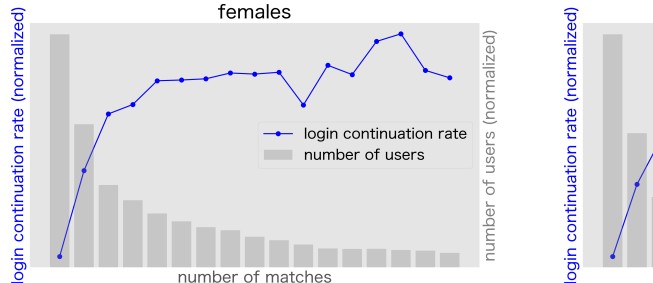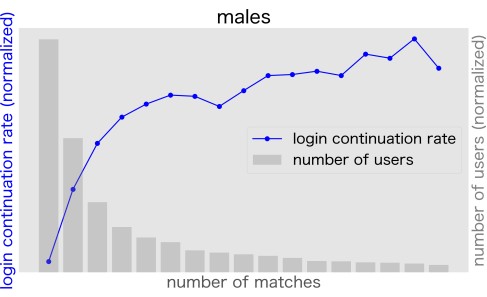

Figure 1: The relationship between retention and the number of matches collected on a large-scale online dating platform, showing females (left) and males (right). Details of the data in Appendix B.

many matches, while many others rarely receive matches (Chen et al., 2023; Celdir et al., 2024). Consequently, users who get insufficient matches can be prompted to leave the platform (Pronk & Denissen, 2020; dos Reis Alba, 2020; Dechant et al., 2019).

Figure 1 shows the probability of a user staying on the platform in the next month given the number of matches they had, collected on a large-scale online dating platform (details of the data setup in Appendix B). We observe that the users with smaller numbers of matches have much higher chances of leaving the platform in the next month. It is problematic for many two-sided platforms to have users leaving the platform, particularly since many of these platforms rely on models where user retention directly determines platform success. An illustrative example is an online dating application that is mostly based on a subscription model, where user retention directly corresponds to profit. When the system focuses only on maximizing the number of matches, recommendations become concentrated on a small group of already popular users. This leaves many others with very few opportunities, even though those users are the most likely to leave if they do not get matches. As a result, the overall number of matches may increase, but the imbalance inevitably causes more users to churn (Chen et al., 2023; Celdir et al., 2024).

To account for the problem of match maximization, one might consider fairness as a potential solution (Tomita & Yokoyama, 2024; Chen et al., 2023; Celdir et al., 2024; Morik et al., 2020). Fairness objectives, such as fairness of exposure (Singh & Joachims, 2018), typically aim to ensure that every user receives the amount of exposure in proportion to their potential utility. However, while fairness might, in rare cases, be the ultimate goal of a platform, this is not true for the majority. Users do not suddenly change their behavior or reward the platform simply because they receive the amount of exposure specified by a fairness objective. In contrast, user retention is the ultimate goal for the majority of platforms, as it is directly tied to long-term sustainability and revenue generation in various business models (e.g., subscription-based services). In real-world practice, where user retention is the ultimate goal, casually choosing fairness formulations or methods leaves the optimization of retention up to chance. Specifically, fairness objectives are only effective if popular users require high exposure to stay on the platform, and less popular users need less exposure to stay. Such a pattern is not promised, as different users would have different satisfaction levels, often regardless of their popularity.

To deal with the problem that existing methods do not account for user retention, we propose a new problem setting, where the objective is to maximize user retention for users on both sides of the matching platform. Instead of axiomatically defining fairness as in existing methods, we build data-driven algorithms based on formulations aligned with the real-world motivation, where user retention is the ultimate goal. Specifically, we introduce a novel dynamic Learning-to-Rank (LTR) algorithm, called **M**atching for **Ret**ention (**MRet**), which provides recommendations that lead to the highest rate of user retention. MRet first learns a personalized retention curve for each user based on profile and interaction history (like Figure 1 but for each user). Then, for each arriving user, MRet jointly considers two perspectives: the retention probability gains of the user receiving the recommendations, as well as the retention probability gains of the users who are being recommended. This means that the recommended users are chosen not only because they increase the likelihood of the receiver retaining, but also because their own retention can be impacted by the interaction. By explicitly modeling retention probability gains from both sides, our algorithm strategically determines the ideal recommendations that best maximize user retention across the entire platform. MRet adapts over time and naturally directs scarce matches to where they have the biggest retention gain

at each time step. As expected yet crucial, empirical evaluations on synthetic and real-world datasets from a major online dating platform demonstrate that our algorithm maximizes user retention, while traditional methods focus on match counts or fairness heuristics instead. The code is provided in the supplementary material.

## 2 PRELIMINARIES

In this section, we formally define the ranking problem in a two-sided matching setting, clearly highlighting the limitations of current matching algorithms.

Consider two distinct sets of users, denoted by $\mathcal{X}$ and $\mathcal{Y}$, representing the two sides of a matching platform (e.g., men and women on an online dating platform). At each time step $\tau$, a single user arrives from one of the two sides: either $x \sim p(x)$, where $x \in \mathcal{X}$, or $y \sim p(y)$, where $y \in \mathcal{Y}$. The arrival distributions $p(x)$ and $p(y)$ are unknown. Note that $x$ and $y$ also contain user context (user profile + interaction history) that we can fully use for recommendation.

Upon arrival, the platform generates a ranking/sequence of $K$ recommendations from the pool of users in the opposite group. The recommendations may be shown in any format: either as a ranked list or sequentially one by one. Specifically, when user $x \in \mathcal{X}$ arrives at time $\tau$, we select an ordered list of candidates

$$\sigma_\tau = [\sigma_{\tau,1}, \sigma_{\tau,2}, \ldots, \sigma_{\tau,K}], \tag{1}$$

with $\sigma_{\tau,k} \in \mathcal{Y}$ for all $k$. Conversely, if the arriving user is $y \in \mathcal{Y}$, then $\sigma_{\tau,k} \in \mathcal{X}$. We follow a common setting used in online matching platforms (Bapna et al., 2023; Pizzato et al., 2010c; Xia et al., 2015b; Tyson et al., 2016), where the arriving user sees two blocks: (i) all users who have already liked them, and (ii) the fresh top-$K$ recommendations $\sigma_\tau$. After viewing the list, the visitor may like or skip each profile.

Since any one-sided like is always shown to the recipient at their next login, the match probability $r(x, y) \in [0, 1]$ specifically represents the likelihood that both users $x$ and $y$ liked each other, resulting in a match. Although the second user's response can arrive with a delay in practice, the like is guaranteed to be shown, and the result is fully determined. Thus, following most online bandit and LTR models (Li et al., 2010; Joachims et al., 2017), we suppose that we observe the binary match indicators immediately after the recommendation for simplicity. In this work, we take $r(x, y)$ as given (either available or estimated upstream), since our focus is on user retention rather than on estimating match probabilities.

Finally, we may let rank position $k$ receive visibility weight $\alpha_k$ with $1 = \alpha_1 \geq \alpha_2 \geq \cdots \geq \alpha_K \geq 0$. Then the expected number of matches for the receiver $x$ at time $\tau$, given a ranking $\sigma_\tau$, is

$$m_\tau(x) = \sum_{k=1}^{K} \alpha_k \, r(x, \sigma_{\tau,k}). \tag{2}$$

### 2.1 MAXIMIZATION OF MATCHES

Typically, existing recommendation algorithms aim to maximize the immediate expected number of matches. Given a user $x$, a conventional matching maximization method generates the recommendation set by sorting in the order of match probability

$$\sigma_{\text{max match}} = \arg\max_{\sigma_\tau} \sum_{k=1}^{K} \alpha_k \, r(x, \sigma_{\tau,k}) = \arg\operatorname{sort}_{y \in \mathcal{Y}} r(x, y). \tag{3}$$

However, optimizing solely for the highest match probabilities leads to imbalanced matches among users. Some users consistently appear in the recommendations, thereby having many matches. Conversely, other users rarely appear high enough in rankings, resulting in limited matches, frustration, and eventually abandonment from the platform (Tomita & Yokoyama, 2024). We observe a clear pattern in Figure 1, that users are more likely to leave the platform with fewer matches. Especially, we observe a special pattern that the first few matches impact retention probabilities much more than increased match counts. Consequently, recommendations based purely on maximizing matches can lead to user abandonment, which is the ultimate objective of the platform.

## 2.2 FAIRNESS IN RANKINGS

Fairness objectives are often used to ensure that even unpopular users receive exposure, which seemingly addresses this problem. The most common fairness objective is the fairness of exposure in a ranking (Morik et al., 2020; Singh & Joachims, 2018).

Specifically, Morik et al. (2020) has introduced a dynamic LTR algorithm that can effectively ensure that each user receives exposure in proportion to their merit. If the arriving user is $x \in \mathcal{X}$, we aim to achieve fair exposure for any candidate user $y \in \mathcal{Y}$.

Let the exposure of candidate $y$ at time step $\tau$ be

$$\text{Exp}_\tau(y) = \sum_{k=1}^{K} \alpha_k \, \mathbb{1}(\sigma_{\tau,k} = y). \tag{4}$$

Then, for any two candidates $y_i, y_j \in \mathcal{Y}$, the disparity

$$D_\tau(y_i, y_j) \;=\; \frac{\frac{1}{\tau} \sum_{t=1}^{\tau} \text{Exp}_t(y_i)}{\mathbb{E}_{p(x)}[r(x, y_i)]} - \frac{\frac{1}{\tau} \sum_{t=1}^{\tau} \text{Exp}_t(y_j)}{\mathbb{E}_{p(x)}[r(x, y_j)]} \tag{5}$$

measures how far amortized exposure over $\tau$ time steps was fulfilled. When the disparity is zero, it means that all users are exposed fairly in relative to their expected utility $\mathbb{E}_{p(x)}[r(x, y)]$, which denotes the expected match probability of user $y$ over the population of users $x$.

Now, to convert the current fairness disparity into an adjustment term, for every candidate $y \in \mathcal{Y}$, we set

$$\text{err}_\tau(y) = (\tau - 1) \max_{y' \in \mathcal{Y}} D_{\tau-1}(y', y). \tag{6}$$

This term is zero for users who already have the highest exposure-merit ratio, and grows linearly in $\tau$ for under-exposed users, ensuring stronger corrections when unfairness persists.

With a trade-off parameter $\lambda > 0$, at time step $\tau$, FairCo builds the rankings by solving

$$\sigma_{\text{FairCo}} = \underset{y \in \mathcal{Y}}{\arg \text{sort}} \left[ r(x_\tau, y) + \lambda \, \text{err}_\tau(y) \right], \tag{7}$$

where the first term prioritizes estimated relevance, and the second lifts candidates from underserved groups.

However, while such fairness objectives can prevent extreme imbalances, they do not directly align with the true objectives of most platforms. Users do not necessarily remain active or reward the system simply because exposure meets the fair target. In practice, user retention is often the more fundamental goal, as it determines both long-term sustainability and revenue. Relying only on fairness, therefore, makes improvements in retention a matter of luck. Specifically, fairness helps only when the required exposure happens to match the retention needs of both popular and less popular users. Since retention behavior varies widely across individuals, such alignment cannot be assumed. As a result, fairness serves as, at best, a heuristic proxy for retention rather than a reliable optimization target.

## 3 OPTIMIZING USER RETENTION

Despite many two-sided platforms having their revenue dependent on user retention, existing methods do not account for user abandonment. To solve the issue, **we define a new problem setting of explicitly optimizing user retention**. User retention is often the ultimate platform objective, as many two-sided matching platforms are reliant on subscriptions. Thus, rather than maximizing matches or targeting match maximization or some heuristic fairness objective, we align the formulation with the real-world motivation, where user retention is the ultimate objective.

To solve this new problem, we come up with a novel dynamic LTR framework that updates the recommendations dynamically given the observed rewards. The framework not only considers the retention of users receiving the recommendations, but also considers the retention of users being recommended.

We first consider a retention function $f(x, m)$ that represents the likelihood that a user $x \in \mathcal{X}$ stays on the platform (i.e., retention probability) given $m$ matches. We can similarly write $f(y, m)$ to represent the reward function of user $y \in \mathcal{Y}$. The reward function $f$ is estimated by simply performing reward regression, using the user context. Now, to derive the optimal ranking, denote the number of matches user $x$ received until time step $\tau$ as

$$m_{1:\tau}(x) = \sum_{t=1}^{\tau-1} m_t(x). \tag{8}$$

At the end of step $\tau$, user $x$ stays with probability $f\big(x, m_{1:\tau}(x) + m_\tau(x)\big)$ and never return.

We aim to recommend a set of users $\sigma$ that would maximize the retention probability gain for both the receiving user $x \in \mathcal{X}$ and the recommended users $\sigma \in \mathcal{Y}^K$. Specifically, given some receiving user $x \in \mathcal{X}$, we propose that the optimal ranking on time step $\tau$ is

$$\sigma_\tau^* = \arg\max_{\sigma_\tau} \bigg\{ \underbrace{f\Big(x, \, m_{1:\tau}(x) + \sum_{k=1}^{K} \alpha_k r\big(x, \sigma_{\tau,k}\big)\Big) - f\big(x, m_{1:\tau}(x)\big)}_{\text{Gain for receiver } x}$$
$$+ \underbrace{\sum_{k=1}^{K} \Big[ f\Big(\sigma_{\tau,k}, \, m_{1:\tau}(\sigma_{\tau,k}) + \alpha_k r\big(\sigma_{\tau,k}, x\big)\Big) - f\big(\sigma_{\tau,k}, m_{1:\tau}(\sigma_{\tau,k})\big) \Big]}_{\text{Gain for recommended users}} \bigg\}. \tag{9}$$

Eq. (9) explicitly chooses a ranking where both the receiver and the recommended users increase their retention probabilities the most.

Figure 2 demonstrates a toy example of how we select the top-1 ranking for Receiver. Here, Candidate A has the most gain on the reward function $f$ from the candidate's point of view, and Receiver will gain the most by selecting Candidate B. However, since we seek to maximize the *total* gain in the reward function, the best option is Candidate C with a total gain of $60\%$, instead of Candidates A or B with the gains of $50\%$.

However, directly solving the optimization problem Eq. (9) is NP-hard. Thus, its direct optimization is not practical.

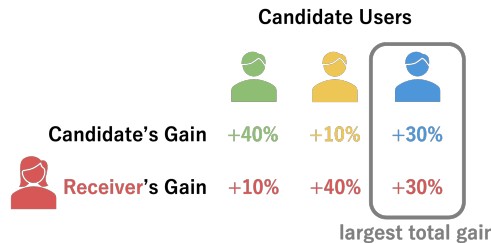

Figure 2: Toy example of how a top-1 ranking would be selected for Receiver. In this case, the optimal recommendation is Candidate C because it raises the *total retention probability* the most.

### 3.1 THE MRET RANKER

To overcome this NP-hard optimization, we introduce our **M**atching for **Ret**ention (**MRet**) ranker. MRet quickly approximates Eq. (9) with theoretical grounding.

To demonstrate MRet, we introduce a realistic assumption: the reward function is concave.

---

**Assumption 1** (Concavity of the Reward Function). *For every user $x \in \mathcal{X}$, the reward function $f(x, \cdot)$ is concave in the number of matches $m \geq 0$. Formally, for all $m_1, m_2 \geq 0$ and every $\theta \in [0, 1]$,*

$$f\big(x, \theta m_1 + (1 - \theta)m_2\big) \geq \theta f(x, m_1) + (1 - \theta) f(x, m_2). \tag{10}$$

*We assume similarly for $y \in \mathcal{Y}$ and $f(y, \cdot)$.*

---

Intuitively, this assumption implies that as users receive more matches, their retention gain is decelerated. This may lead to a peak retention probability at an optimal number of matches, beyond which additional matches no longer help the retention probability. For example, a user who gets their first match gains more incremental retention probability than when they get the 10th match. This is a

very realistic assumption, as we can also observe this pattern in Figure 1, which is collected on an actual online dating application. We also provide experimental results where this assumption does not hold.

Under Assumption 1, the receiver-side term in Eq. (9) admits a Jensen-type lower bound. Writing $A = \sum_{j=1}^{K} \alpha_j$, concavity of $f(x, \cdot)$ gives the following lemma.

**Lemma 1** (Jensen lower bound). *Under the Assumption 1, for a ranking $\sigma_\tau$,*

$$f\left(x, m_{1:\tau}(x) + \sum_{k=1}^{K} \alpha_k r(x, \sigma_{\tau,k})\right) \geq \sum_{k=1}^{K} \frac{\alpha_k}{A} f\left(x, \ m_{1:\tau}(x) + Ar(x, \sigma_{\tau,k})\right). \quad (11)$$

On the candidate side, the contribution of a user $y$ depends on the position-specific visibility weight $\alpha_k$. Concavity again provides a way to bound this dependence using a linear factor based on the maximum weight $\alpha_{\max}$.

**Lemma 2** (Concavity-based linear bound). *Under Assumption 1, for any candidate $y \in \mathcal{Y}$ and any $0 \leq \alpha_k \leq \alpha_{\max}$,*

$$f\big(y, m_{1:\tau}(y) + \alpha_k r(x, y)\big) - f\big(y, m_{1:\tau}(y)\big)$$
$$\geq \frac{\alpha_k}{\alpha_{\max}} \Big( f\big(y, m_{1:\tau}(y) + \alpha_{\max} r(x, y)\big) - f\big(y, m_{1:\tau}(y)\big)\Big).$$

We prove the Lemmas 1 and 2 in Appendix C.1 and C.2. Using Lemma 1 and 2, we reformulate the original optimization problem Eq. (9) into a tractable lower bound maximization problem. In particular, Lemma 1 provides a decomposition of the receiver side, and by further applying a concavity-based linear lower bound to each candidate user, the objective can be expressed as

$$\max_{\sigma_\tau} \sum_{k=1}^{K} \alpha_k \, \text{Score}(\sigma_{\tau,k}), \quad (12)$$

where, with $\alpha_{\max} = \max_k \alpha_k$, the per-item score is defined as

$$\text{Score}(y) = \frac{1}{A} \, f(x, m_{1:\tau}(x) + Ar(x, y))$$
$$+ \frac{1}{\alpha_{\max}} \left[ f(y, m_{1:\tau}(y) + \alpha_{\max} r(x, y)) - f(y, m_{1:\tau}(y)) \right]. \quad (13)$$

By the rearrangement inequality, assigning higher-score candidates to positions with larger visibility weights $\alpha_k$ maximizes this lower bound. Therefore, the MRet ranking is obtained as

$$\sigma_{\text{MRet}} = \underset{y \in \mathcal{Y}}{\mathbf{arg\ sort}} \, \text{Score}(y). \quad (14)$$

This allows us to solve the problem using argsort, significantly reducing computational complexity. Specifically, instead of an NP-hard problem, it is transformed into one where we compute an individual score for each potential candidate user from the opposing set. Assuming constant-time function evaluations and lookups, the operation takes $O(N \log N)$ time. This is a substantial improvement over the complexity of Eq. (9). We now provide its empirical performance through extensive experiments.

## 4 EXPERIMENTS

In this section, we report the experimental results with synthetic data and real-world data from a Japanese online dating platform. Our experiment code is available at https://github.com/kishi6/ICLR2026_MRet.

## 4.1 SYNTHETIC EXPERIMENTS

To generate synthetic data, we consider two user groups, $\mathcal{X}$ and $\mathcal{Y}$. We sample 10-dimensional context vectors $x \in \mathcal{X}, y \in \mathcal{Y}$ from the standard normal distribution. We synthesize the match probability between user $x$ and user $y$ as

$$r(x,y) = (1 - \kappa) \cdot r_{base}(x,y) + \kappa \cdot r_{pop}(x,y),$$

where $r_{base}(x,y) := \frac{x \cdot y}{||x|| ||y||}$ represents the base reward using cosine similarity, which captures the compatibility between user $x$ and user $y$. The second term introduces a popularity skew, where $r_{pop}(x,y) = pop_x \cdot pop_y$ and $pop_x, pop_y$ are sampled from a uniform distribution with range $[0,1]$. $pop_x$ and $pop_y$ represent the popularity levels of user $x$ and user $y$, respectively. The parameter $\kappa \in [0,1]$ controls the strength of the popularity skew, and a larger value of $\kappa$ leads to a stronger bias. We assume that $r(x,y)$ is known in advance when selecting a ranking. Next, the user retention label $u \in \{0,1\}$ is sampled from a Bernoulli distribution with the user retention probability determined by $f(x, m_{1:\tau}(x))$, where $m_{1:\tau}(x)$ denotes the cumulative number of matches until time step $\tau$ for user $x$. We define the user retention probability $f$ as

$$f(x, m_{1:\tau}(x)) = \begin{cases} a_x \left(m_{1:\tau}(x) - b_x\right)^2 + 0.95 & (0 \leq m_{1:\tau}(x) \leq b_x) \\ 1 - 0.05 \cdot \exp\left(2(b_x - m_{1:\tau}(x))\right) & (m_{1:\tau}(x) > b_x) \end{cases} \quad (15)$$

where $a_x = x \cdot M_a$ [1] and $b_x = x \cdot M_b$ are scalar values determined by the user feature vector $x$. We refer to $b_x$ as the *satisfactory match count*, since it represents the number of matches at which the user retention probability substantially levels off. The probability increases quadratically with the cumulative number of matches $m_{1:\tau}(x)$ until reaching $b_x$, after which the growth becomes gradual.

We simulate the dynamics of user retention over time. Let $\mathcal{X}_{churn}^\tau$ denote the set of users who have left the platform by time $\tau$, and $\mathcal{X}_{retention}^\tau$ denote the set of users who remain on the platform. At each time step, we randomly select a user group ($\mathcal{X}$ or $\mathcal{Y}$) according to a proportion parameter $\rho \in [0,1]$. From the selected group, we then sample a user uniformly at random from those who remain on the platform (e.g., if $\mathcal{X}$ is selected, we sample a user $x \in \mathcal{X}_{retention}^\tau$). According to each method, the algorithm recommends a ranked list of users from the opposite group (e.g., $y \in \mathcal{Y}_{retention}^\tau$), and the cumulative number of matches $m_{1:\tau}$ is updated based on the ranking and $r(x,y)$. After the recommendation phase, we perform a retention phase in which we independently select every user (including those not involved in the ranking process) with a probability of 0.2% to undergo a churn evaluation. For each selected user, we sample a user retention label $u \in \{0,1\}$ from a Bernoulli distribution parameterized by $f(\cdot, m_{1:\tau}(\cdot))$. If $u = 0$, the user is removed from the platform and $\mathcal{X}_{retention}^\tau$ and $\mathcal{Y}_{retention}^\tau$ are updated accordingly. We repeat this process from time step $\tau = 0$ to $\tau = T$.

**Compared methods.** We compare MRet with Max Match, Uniform, and FairCo (Morik et al., 2020). FairCo's fairness weight $\lambda$ is set to 100, but we ablate and confirm in Appendix D that different hyperparameters do not change the conclusion. Uniform selects random rankings without considering matches. MRet trains a regression model $\hat{f}$ using XGBoost Chen & Guestrin (2016) on a dataset $\{x, m, u\}_{i=1}^n$. We synthesize this dataset by generating $n$ samples, each consisting of a user feature vector $x$, a match count $m$ drawn from an exponential distribution with mean 2.0, and a retention label $u$ determined according to the retention function $f$. For reference, we also report the result of **MRet (best)**, which has access to the ground-truth function $f$ and thus serves as an oracle upper bound of MRet. We clarify here that Tomita & Yokoyama (2024) cannot be compared as a baseline, because they calculate a doubly-stochastic matrix which takes exponential time when done in a dynamic LTR setting.

**Results.** We use default parameters of $|\mathcal{X}| = 1000$, $|\mathcal{Y}| = 1000$, $K = 5$, $n = 5000$, $T = 2000$, $\kappa = 0.5$, $\rho = 0.5$ and $\alpha_k = 1/k$. Each experiment is repeated 10 times with different random seeds, and we report the average results. We show cumulative number of matches per user $\left( \frac{\sum_{x \in \mathcal{X}} m_{1:T}(x) + \sum_{y \in \mathcal{Y}} m_{1:T}(y)}{|\mathcal{X}| + |\mathcal{Y}|} \right)$ and user retention rate $\left( \frac{|\mathcal{X}_{retention}^T| + |\mathcal{Y}_{retention}^T|}{|\mathcal{X}| + |\mathcal{Y}|} \right)$.

---

[1] Note that $a_x$ is scaled to take negative values so that the user retention function $f$ becomes concave.

**How does MRet perform as the time step increases?**    Figure 3 shows how the cumulative number of matches per user and user retention rate change as the time step $\tau$ increases. As $\tau$ increases, the cumulative number of matches per user grows approximately proportionally across all methods. Max Match achieves the highest number of matches by a significant margin, whereas Uniform results in the lowest. Despite the substantial gap in total matches, Uniform and MaxMatch achieve almost the same user retention rate. This indicates that simply maximizing the number of matches does not necessarily help reduce user drop-off. MRet achieves a higher user retention rate than all baselines, including FairCo, Max Match, and Uniform, with only about 70% of the matches obtained by Max Match. While MRet does not reach the performance of MRet (best), it maintains a high level of user retention by effectively allocating matches.

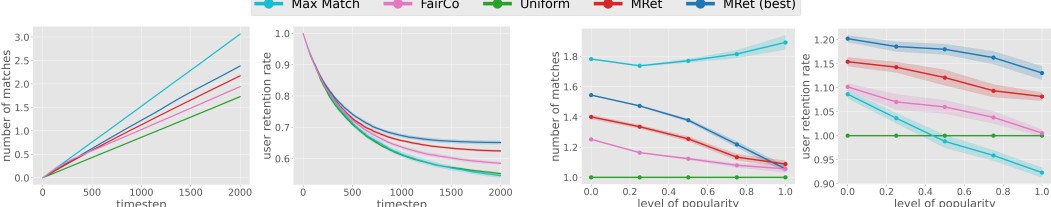

Figure 3: Comparison of cumulative number of matches and user retention rate across different time steps ($\tau$).

Figure 4: Comparison of cumulative number of matches and user retention rate (normalized by Uniform) under varying levels of user popularity ($\kappa$).

**How does MRet perform when user popularity varies?**    Next, we evaluate how MRet performs under varying levels of user popularity $\kappa$, as shown in Figure 4. We adjust the parameter $\kappa$ to control the degree of popularity imbalance across users. When popularity is highly skewed, it becomes difficult for any method to allocate matches to less popular users. As a result, the gap in user retention rate between each method and Uniform tends to narrow. Max Match tends to achieve higher match counts as popularity skew increases. However, user retention decreases significantly because popular users receive an even greater share of the matches. FairCo, which emphasizes fairness, often ends up allocating more matches to already popular users under skewed settings. This leads to increased disparity and ultimately results in performance that is comparable to that of Uniform. MRet maintains a high user retention rate across varying levels of popularity and is particularly effective in preventing retention loss under severe popularity imbalance.

**Why does Fairco underperform in user retention?**    Here, we clarify the reason why FairCo performs worse in terms of user retention, although the cumulative number of matches it achieves is not substantially different from that achieved by MRet (best). Figure 5a shows the histogram of actual match counts among users who remain on the platform at time $T$. While FairCo and MRet (best) differ slightly in the total number of matches, their distributional shapes are similar. Nevertheless, Figure 5b reveals a critical difference between FairCo and MRet (best). This figure shows a histogram of the differences between each user's actual match counts and their satisfactory match counts ($m_{1:T} - b$). Since $b$ represents the threshold at which the retention function levels off, the quantity $m_{1:T} - b$ directly indicates whether a user has obtained fewer or

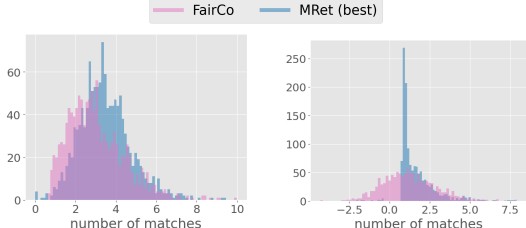

(a) Histogram of actual match counts $m_{1:T}$.

(b) Histogram of deviations between actual and satisfactory match counts $m_{1:T} - b$.

Figure 5: Comparison of match allocation strategies between FairCo and MRet for remaining users at time step $T$.

more matches than this satisfactory level. FairCo allocates matches without considering user retention, resulting in a distribution of match counts that is spread out like a normal distribution, regardless of each user's satisfactory match counts. In contrast, MRet yields mostly non-negative

differences, indicating that it successfully identifies and allocates matches according to each user's necessary amount. These results suggest that fair methods are not sufficient to prevent user churn.

Appendix D further reports and discusses (i) the performance with varying numbers of users, (ii) the performance on different FairCo hyperparams, (iii) the performance under varying noise levels in the match probabilities, (iv) the performance when popularity drifts over time, (v) the performance when FairCo uses the equal-exposure fairness criterion, and (vi) comparison of MRet with the optimal NP-hard recommendations.

## 4.2 REAL-WORLD EXPERIMENTS

To assess the real-world applicability of MRet, we report the results of real-world experiments. We use interaction data from a large-scale Japanese online dating platform with millions of registered users. On this platform, male users receive recommendation lists of female users and can either send a "like" or skip each recommendation. When a male user sends a "like," the female user receives a notification and either matches with him or declines the interaction. Once they form a match, the users can initiate a conversation. Female users follow a similar mechanism in the reverse direction.

To construct the dataset, we first select 1,000 male users and 1,000 female users with relatively many interactions, and create a $1000 \times 1000$ matching matrix. Each element of this matrix represents the matching probability, denoted $r(x, y)$, which takes the value 1 if user $x$ and user $y$ have matched and 0 otherwise. For user pairs with unobserved interactions, we apply the Alternating Least Squares (ALS) Hu et al. (2008) algorithm to impute missing values. We also define user retention as whether the user logs in again in the following month. Specifically, among users who logged in during February 2025, we label those who also log in March 2025 as retained, and those who do not as churned. We aggregate the cumulative match counts for the user retention probability function from matches obtained in February 2025. Since we cannot obtain user retention labels for arbitrary numbers of matches for the users included in the matching matrix, we construct the user retention probability function $f$ using a separate dataset of 60,000 records containing user features, cumulative match counts, and corresponding user retention labels.

To create the user retention probability $f$, we first apply k-means clustering to partition male and female users separately into five clusters based on their user features. For each cluster, we calculate the average user retention label at each level of cumulative match count. We assume that users within the same cluster share similar retention behavior, and define the function $f$ by assigning the corresponding average retention label to each combination of cluster and match count. This enables us to assign the retention probability for any user based on their cluster and the cumulative number of matches. Using these definitions derived from real data, the simulation follows the same procedure as the synthetic data experiments. The detailed experimental settings are provided in Appendix E.

**Results** We evaluate MRet against Max Match, FairCo, and Uniform varying time steps $\tau$ on the real-world data in Figure 6. In contrast to the synthetic experiments, most elements in the matching matrix are zero in the real-world data setting, indicating that matches rarely occur. Under such sparse conditions, fairness-oriented methods such as FairCo and Uniform fail to perform effective matching, resulting in significantly lower user retention. Max Match turns out to be relatively effective, achieving higher retention than the fairness-based baselines. Notably, MRet achieves the highest user retention with extremely sparse match data. Moreover, although MRet is origi-

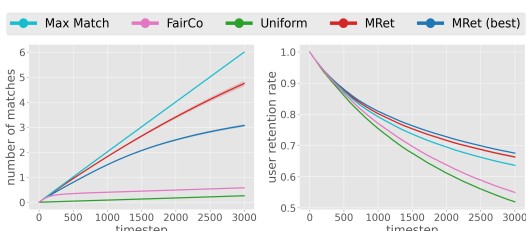

Figure 6: Comparison of cumulative number of matches and user retention rate across different time steps ($\tau$) on real-world data.

nally designed under the assumption 1 that the user retention function $f$ is concave in the number of matches, it still performs well even when applied to non-concave functions in the real-world data experiments. This demonstrates that MRet is not only effective but also robust and broadly applicable in practical scenarios.

Appendix E further reports and discusses the performance (i) with different training data sizes, (ii) under different exposure probabilities, (iii) under different group recommendation ratios and (iv) under large-scale user setting.

## 5 RELATED WORK

Two-sided matching platforms, such as online dating (Neve & Palomares, 2019; Pizzato et al., 2010b; Xia et al., 2015a) and recruitment (Jiang et al., 2020; Le et al., 2019; Yang et al., 2022), differ from conventional recommender systems in that they recommend users from one side of the platform to those on the other, rather than recommending items to users. This reciprocal nature has motivated the development of Reciprocal Recommender Systems (RRS), which typically focus on maximizing the total number of matches (Mine et al., 2013; Pizzato et al., 2010a; Palomares et al., 2021; Pizzato et al., 2010b; Qu et al., 2018; Su et al., 2022; Hayashi et al., 2025). However, this objective often results in severe imbalances, where popular users accumulate a large number of matches, whereas many others receive very few (Chen et al., 2023; Celdir et al., 2024). Such disparities can drive under-served users to leave the platform (Pronk & Denissen, 2020; dos Reis Alba, 2020; Dechant et al., 2019), ultimately threatening platform sustainability.

To address these imbalances, fairness-oriented approaches have been widely studied (Singh & Joachims, 2018; Tomita & Yokoyama, 2024; Tomita et al., 2023; Celdir et al., 2024; Morik et al., 2020; Devic et al., 2023; Saito & Joachims, 2022). Several works explicitly integrate fairness into two-sided matching (Tomita & Yokoyama, 2024; Do et al., 2021; Xia et al., 2019). In addition, fairness-aware dynamic Learning-to-Rank (LTR) algorithms have been developed, where past feedback influences future rankings to ensure that users receive exposure proportional to their merit (Yang & Ai, 2021; Morik et al., 2020; Wang et al., 2024; Biswas et al., 2021). Although these approaches mitigate disparities, fairness serves only as a heuristic principle rather than a direct objective, and it does not necessarily align with maximizing user satisfaction or retention.

Beyond short-term optimization, recommender systems research has increasingly emphasized long-term user engagement. Several studies have proposed models that optimize return visits or subscription renewals (Wu et al., 2017; Zou et al., 2019; McDonald et al., 2023; Hohnhold et al., 2015; Takehi et al., 2025; Saito et al., 2024) highlighted the necessity of incorporating long-term satisfaction into search and recommendation models. These works underscore the importance of long-term user engagement, which constitutes a primary objective for platforms.

We propose a new problem setting, where the objective is to maximize user retention on both sides of the platform. Most prior approaches in two-sided matching optimize probabilistic recommendation lists represented as doubly stochastic matrices (Su et al., 2022; Tomita & Yokoyama, 2024; Tomita et al., 2023). This formulation becomes computationally prohibitive as the number of users grows. In contrast, we introduce a different problem setting that extends dynamic LTR (Morik et al., 2020) to two-sided matching. This framework more realistically captures sequential user arrivals, significantly reduces computational complexity, and enables direct optimization of user retention.

## 6 CONCLUSIONS

This paper introduces a new problem of optimizing user retention in a two-sided matching platform. While existing studies on two-sided platforms typically aim to maximize the number of successful matches between the two user sides, this makes some users receive more matches than others, prompting unpopular users to leave the platform. A high user abandonment rate is often unacceptable for many two-sided matching platforms, as long-term sustainability and revenue depend on retaining users. Fairness objectives have been explored as a way to mitigate imbalance, but they are not the ultimate goal for most platforms and do not reliably ensure retention. Therefore, instead of axiomatically defining fairness, we formulate user retention as the main objective, directly reflecting the real-world motivation. We provide a theoretically grounded dynamic learning-to-rank algorithm called MRet, with empirical evaluations on synthetic data and real-world data collected on a major online-dating platform. As naturally expected but important, our results demonstrate that MRet retains significantly more users than match maximization and fairness-based methods.

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

## A    Limitations & Future Work

While our work introduces a novel problem setting that directly maximizes user retention in two-sided matching, several limitations remain. Our model assumes that the matching matrix, which encodes the preferences of both sides, is given in advance. In practice, such preferences need to be estimated either offline or online, and an important extension of our framework is to incorporate this estimation process directly into the model. We also learn user retention functions from offline data, whereas in real-world platforms, it would be valuable to adaptively update them online, potentially leading to new models that balance exploration and exploitation. Moreover, our current formulation treats both sides of the platform as equally important, while in practice the platform's business model may prioritize one side over the other. For instance, in online dating, male users are often the paying side and retention of that group may be more critical, whereas in job matching, where employer demand often exceeds job seeker supply, it may be desirable to place greater weight on the retention of job seekers.

## B    Detailed Description of Figure 1

We provide a detailed description of the data aggregation procedure in Figure 1. The figure shows the relationship between the number of matches obtained by users and their login continuation rate based on data collected from a large-scale online dating platform in Japan. We focus on users who logged in during February 2025, where login continuation is defined as whether the user logged in again in March 2025. For each user, we calculate the total number of matches obtained in February 2025. Male and female users are binned separately according to their total number of matches, and for each bin, we report both the average login continuation rate and the number of users it contains. All values are normalized by the bin with the smallest number of matches.

## C    Proof

Here, we provide the derivations and proofs that are omitted in the main text.

### C.1    Proof of Lemma1

$$
f\left(x, m_{1:\tau}(x) + \sum_{k=1}^{K} \alpha_k r(x, \sigma_{\tau,k})\right)
$$

$$
= f\left(x, \sum_{k=1}^{K} \frac{\alpha_k}{A} m_{1:\tau}(x) + \sum_{k=1}^{K} \frac{A}{A} \alpha_k r(x, \sigma_{\tau,k})\right) \because A = \sum_{k=1}^{K} \alpha_k > 0
$$

$$
= f\left(x, \sum_{k=1}^{K} \frac{\alpha_k}{A} \left(m_{1:\tau}(x) + A r(x, \sigma_{\tau,k})\right)\right)
$$

$$
\geq \sum_{k=1}^{K} \frac{\alpha_k}{A} f\left(x, \ m_{1:\tau}(x) + A r(x, \sigma_{\tau,k})\right) \quad \because \text{Jensen's inequality under Assumption 1}, \sum_{k=1}^{K} \frac{\alpha_k}{A} = 1
$$

### C.2    Proof of Lemma2

$$
f(y, m_{1:\tau}(y) + \alpha_k r(y, x)) - f(y, m_{1:\tau}(y))
$$

$$
= f\left(y, \left(1 - \frac{\alpha_k}{\alpha_{max}}\right) m_{1:\tau}(y) + \frac{\alpha_k}{\alpha_{max}} (m_{1:\tau}(y) + \alpha_{max})\right) - f(y, m_{1:\tau}(y)) \because \alpha_{max} = \max_k \alpha_k > 0
$$

$$
\geq \left(1 - \frac{\alpha_k}{\alpha_{max}}\right) f(y, m_{1:\tau}(y)) + \frac{\alpha_k}{\alpha_{max}} f(y, m_{1:\tau}(y) + \alpha_{max}) - f(y, m_{1:\tau}(y)) \because \text{Assumption1}
$$

$$
= \frac{\alpha_k}{\alpha_{\max}} \left(f(y, m_{1:\tau}(y) + \alpha_{\max} r(y, x)) - f(y, m_{1:\tau}(y))\right)
$$

### C.3 PROOF THAT MRET MAXIMIZES A LOWER BOUND OF THE OPTIMIZATION PROBLEM

We show that the original optimization problem Eq. (9) can be reformulated as a tractable lower bound maximization problem Eq. (14).

$$f\Big(x,\, m_{1:\tau}(x) + \sum_{k=1}^{K} \alpha_k r\big(x, \sigma_{\tau,k}\big)\Big) - f\big(x, m_{1:\tau}(x)\big)$$

$$+ \sum_{k=1}^{K} \Big[ f\Big(\sigma_{\tau,k},\, m_{1:\tau}(\sigma_{\tau,k}) + \alpha_k r\big(\sigma_{\tau,k}, x\big)\Big) - f\big(\sigma_{\tau,k}, m_{1:\tau}(\sigma_{\tau,k})\big) \Big]$$

$$\geq \sum_{k=1}^{K} \frac{\alpha_k}{A}\, f\big(x,\, m_{1:\tau}(x) + A\, r(x, \sigma_{\tau,k})\big) - f\big(x, m_{1:\tau}(x)\big)$$

$$+ \sum_{k=1}^{K} \Big[ f\Big(\sigma_{\tau,k},\, m_{1:\tau}(\sigma_{\tau,k}) + \alpha_k r\big(\sigma_{\tau,k}, x\big)\Big) - f\big(\sigma_{\tau,k}, m_{1:\tau}(\sigma_{\tau,k})\big) \Big] \qquad \because \text{Lemma 1}$$

$$\geq \sum_{k=1}^{K} \frac{\alpha_k}{A}\, f\big(x,\, m_{1:\tau}(x) + A\, r(x, \sigma_{\tau,k})\big) - f\big(x, m_{1:\tau}(x)\big)$$

$$+ \sum_{k=1}^{K} \frac{\alpha_k}{\alpha_{\max}} \Big( f\big(\sigma_{\tau,k},\, m_{1:\tau}(\sigma_{\tau,k}) + \alpha_{\max} r(\sigma_{\tau,k}, x)\big) - f\big(\sigma_{\tau,k}, m_{1:\tau}(\sigma_{\tau,k})\big) \Big) \quad \because \text{Lemma 2}$$

$$= \sum_{k=1}^{K} \alpha_k\, \text{Score}(\sigma_{\tau,k}) - f\big(x, m_{1:\tau}(x)\big),$$

where

$$\text{Score}(\sigma_{\tau,k}) := \frac{1}{A}\, f\big(x,\, m_{1:\tau}(x) + A\, r(x, \sigma_{\tau,k})\big) + \frac{1}{\alpha_{\max}} \Big[ f\big(\sigma_{\tau,k},\, m_{1:\tau}(\sigma_{\tau,k}) + \alpha_{\max} r(\sigma_{\tau,k}, x)\big) - f\big(\sigma_{\tau,k},\, m_{1:\tau}(\sigma_{\tau,k})\big) \Big].$$

Since $1 = \alpha_1 \geq \alpha_2 \geq \cdots \geq \alpha_K \geq 0$, it follows that Eq. (14) ensures that MRet maximizes a lower bound of the original optimization problem.

## D SYNTHETIC EXPERIMENTS

**Additional Results.** We report and discuss additional synthetic experiment results.

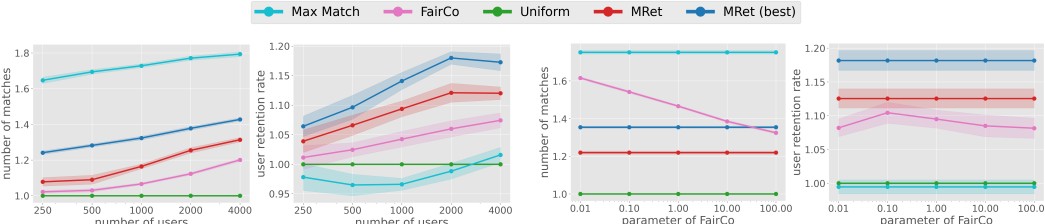

Figure 7: Comparison of cumulative number of matches and user retention rate (normalized by Uniform) under varying the number of users $(|\mathcal{X}| + |\mathcal{Y}|)$.

Figure 8: Comparison of cumulative number of matches and user retention rate (normalized by Uniform) under varying the hyperparameter of FairCo $(\lambda)$.

**How does the proposed method perform when varying the number of users?** Figure 7 shows the results when increasing the number of users $(|\mathcal{X}| + |\mathcal{Y}|)$ with each value normalized by the result of Uniform. As the number of users increases, the number of possible recommendation candidates also increases, and thus all methods show improvements in both the cumulative number of matches

and user retention rate compared to Uniform. Max Match slightly outperforms Uniform only when the number of users is large. In contrast, MRet consistently exhibits superior performance across all user sizes, with a significant margin.

**How does the hyperparameter of FairCo affect its performance?** Figure 8 shows the experimental results when varying the hyperparameter of FairCo $\lambda$ in Equation 7. A smaller value of $\lambda$ places more emphasis on maximizing the number of matches, while a larger value emphasizes fairness. As expected, increasing $\lambda$ results in a decrease in the number of matches, which aligns with the theoretical perspective. Regarding user retention rate, FairCo achieves the best user retention rate at $\lambda = 0.1$, but still performs worse than MRet. This shows that even with careful tuning, fair metrics cannot outperform MRet. In contrast, MRet requires no parameter tuning, making it easier to apply in real-world scenarios.

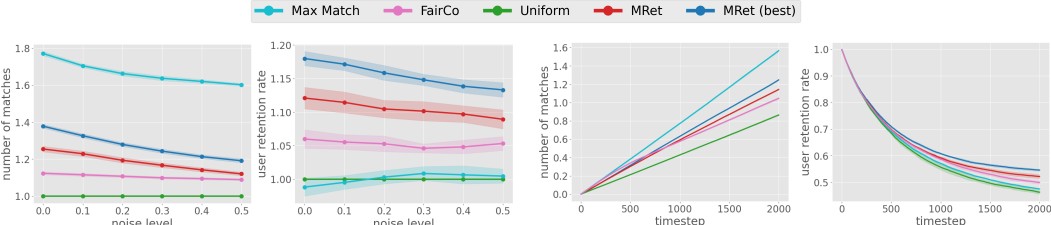

Figure 9: Comparison of cumulative number of matches and user retention rate (normalized by Uniform) under varying noise levels in the match probabilities ($\delta$).

Figure 10: Comparison of cumulative number of matches and user retention rate (normalized by Uniform) when popularity drifts over time.

**How does MRet perform under varying noise levels in the match probabilities?** To empirically assess the robustness of MRet to match probabilities, we conducted an experiment in which noise was injected into the match probabilities, as shown in Figure 15. Specifically, we perturb the true probabilities according to

$$\tilde{r}(x, y) = r(x, y) + \varepsilon_{x,y}, \qquad \varepsilon_{x,y} \sim \mathcal{U}[-\delta, \delta],$$

where $\delta$ controls the noise level. The results show that as the noise level increases and the accuracy of the match probabilities deteriorates, the number of matches decreases for MaxMatch, MRet, and MRet (best). Moreover, both MRet and MRet (best) exhibit lower user retention rates as the noise increases. However, across all noise levels, MRet consistently outperforms all baseline methods, demonstrating that it remains robust even when the match probabilities are substantially perturbed.

**How does MRet perform when popularity drifts over time?** We conducted an experiment to test how MRet adapts when popularity drifts over time, as shown in Figure 10. In this experiment, we introduce a time-dependent popularity skew, defining the match probability at time step $t$ as

$$r_t(x, y) = (1 - \kappa)\, r_{\text{base}}(x, y) + \kappa\, pop_t(x)\, pop_t(y).$$

We assign each user $x \in \mathcal{X}$ and $y \in \mathcal{Y}$ one of three popularity trajectories: constant, increasing, or decreasing. For each user $i$, we sample an initial popularity $c_i \sim \text{Uniform}(0, 1)$ and set a fixed slope of 0.5. Letting $\tau_t \in [0, 1]$ denote the normalized time step index, we define the time-varying popularity as

$$pop_t(i) = \begin{cases} c_i, & \text{constant}, \\ c_i + 0.5\tau_t, & \text{increasing}, \\ c_i - 0.5\tau_t, & \text{decreasing}. \end{cases}$$

We then clip the resulting $pop_t(i)$ to the range $[0, 1]$ to ensure it remains within a valid bound. Figure 10 shows that MRet outperforms all baselines in terms of user retention rate, demonstrating its effective adaptation to popularity drifts. This confirms that MRet successfully adapts to popularity drifts, as expected from its dynamic design.

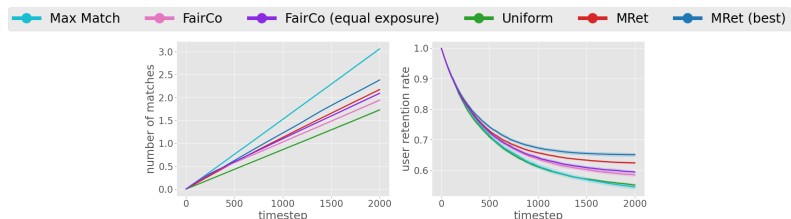

Figure 11: Comparison of cumulative number of matches and user retention rate when FairCo uses the equal-exposure fairness criterion.

**How does FairCo perform under the equal-exposure fairness criterion?** Figure 11 shows the results from an experiment where we modify FairCo's fairness definition to require *equal exposure* across all users, using the same experimental setting as in Figure 3. We implement this variant by removing the "merit" component from FairCo's original objective. Now, the disparity function for FairCo (equal exposure) is given by

$$D_\tau(y_i, y_j) = \frac{1}{\tau} \sum_{t=1}^{\tau} \text{Exp}_t(y_i) - \frac{1}{\tau} \sum_{t=1}^{\tau} \text{Exp}_t(y_j).$$

FairCo (equal exposure) achieves slightly better performance than the original FairCo, but it follows the same overall trend and still performs worse than MRet.

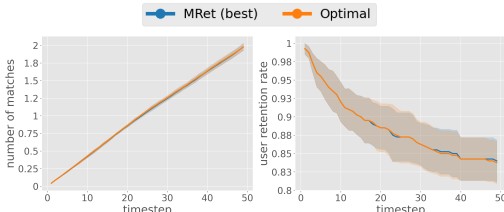

Figure 12: Comparison of cumulative number of matches and user retention rate between FairCo and MRet across different time steps ($\tau$).

**How accurate is MRet as an approximation to the Optimal method?** Figure 8 compares the performance of MRet with that of the Optimal method, which directly maximizes the ranking objective defined in Eq. 9. Since Optimal requires exhaustive search over all possible rankings to select the best one, it cannot be applied to large-scale settings due to computational constraints. We therefore conduct experiments in a small-scale setting with $|\mathcal{X}| = 20$, $|\mathcal{Y}| = 20$, $K = 3$, and $T = 50$, and compare the performance of the two methods as the time step progresses. As shown in the figure, the cumulative number of matches and the user retention rate are almost identical. This indicates that although MRet maximizes a lower bound of the objective function rather than the objective itself, it achieves sufficiently high approximation accuracy while drastically reducing computation by only requiring the sorting of item scores.

## E  REAL-WORLD EXPERIMENTS

**Detailed Setup.** We describe the training procedure of $f$ and the data employed in this process. We compute the values of the true user retention probability function $f$ using the cumulative match counts $m$ from the real data and the user clusters $c$ obtained by $k$-means. We then sample the retention label $u$ from a Bernoulli distribution parameterized by $f$. Finally, we train the predictive model $\hat{f}$ with XGBoost using the resulting triples $(c, m, u)$. In practice, we use the XGBClassifier implementation with 200 trees, a maximum depth of 6, and a learning rate of 0.05.

In the real-world experiments, we set the default parameters to $|\mathcal{X}| = 1000$, $|\mathcal{Y}| = 1000$, $K = 5$, $n = 5000$, $T = 3000$, $\rho = 0.5$, and $\alpha_k = 1/k$. Each experiment is repeated 10 times with different random seeds, and we report the average results.

**Additional Results.** We report and discuss additional real-world experiment results.

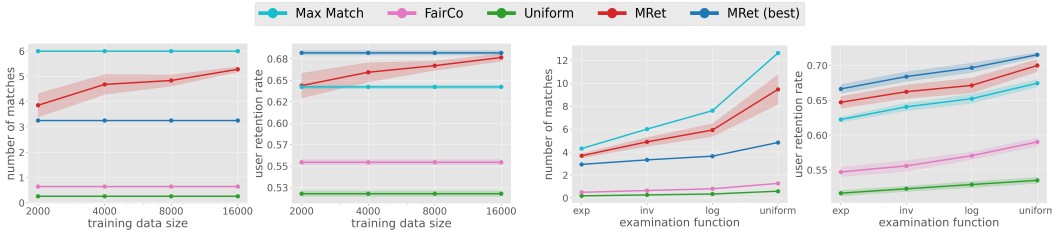

Figure 13: Comparison of cumulative number of matches and user retention rate under varying training data sizes ($n$) on real-world data.

Figure 14: Comparison of cumulative number of matches and user retention rate under varying examination probability functions ($\alpha_k$) on real-world data.

**How does the performance of MRet change as training data size increases?** Figure 13 shows the results when varying the number of training data sizes $n$ used to train the function $\hat{f}$ that predicts user retention probability. As the amount of training data increases, the performance of MRet gradually improves, and when $n = 16000$, it achieves a performance level comparable to MRet (best), which uses the true function $f$. Even in the case where training data is limited ($n = 2000$), MRet significantly outperforms FairCo and Uniform, and performs at least on par with Max Match. These results indicate that MRet is effective even with relatively small amounts of training data.

**How does MRet perform under variations in examination probability?** Next, Figure 14 shows the results under varying examination probability functions. We consider four types of examination functions: 'exp' corresponds to $\alpha_k = 1/\exp(k)$, 'inv' to $\alpha_k = 1/k$, 'log' to $\alpha_k = 1/\log_2(2 + k)$, and 'uniform' to $\alpha_k = 1$. The figure is arranged so that values on the right correspond to more uniform examination probabilities across positions, while values on the left indicate stronger position bias, where higher-ranked positions receive substantially more attention. MRet consistently outperforms existing baseline methods across all examination probability patterns. This confirms that MRet exhibits high robustness to variations in the examination probability.

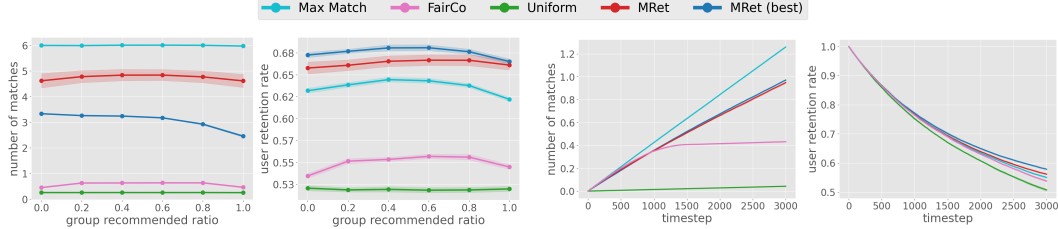

Figure 15: Comparison of cumulative number of matches and user retention rate under varying the ratio of male and female users receiving recommendations ($\rho$) on real-world data.

Figure 16: Comparison of cumulative number of matches and user retention rate under large-scale user settings on real-world data.

**How does MRet perform under variations in group recommendation ratio?** Figure 15 shows the experimental results when varying the ratio $\rho$, which controls whether male or female users receive recommendations on the platform. When $\rho = 0$, the system recommends only male users to female users, whereas when $\rho = 1.0$, it recommends only female users to male users. The results suggest that MRet performs well even under extreme settings where only one side (either male or female) receives recommendations. While this experiment is based on online dating data, it also implies that MRet is applicable to more general matching scenarios, such as job matching, where one side is always the recipient of recommendations ($\rho = 0$ or $\rho = 1.0$). This demonstrates that MRet is broadly applicable to a wide range of matching problems.

**How does MRet perform under large-scale user settings on real-world data?** To examine how MRet performs in large-scale user settings, we conduct an experiment on real data with $n_x = n_y = 5000$ in Figure 16. We observe that even in this substantially larger setting, MRet remains the best-performing algorithm.

