# OpenReview forum: "Beyond Match Maximization and Fairness: Retention-Optimized Two-Sided Matching"
_ICLR.cc/2026/Conference — ICLR 2026 Poster_

### Official Review · Reviewer_WdJu · 2025-10-26

**Soundness:** 3
**Presentation:** 3
**Contribution:** 2
**Rating:** 4
**Confidence:** 2

**Summary:**

In this paper, the authors studied a new paradigm for two-sided matching platforms that optimizes user retention rather than traditional objectives like match maximization or fairness. The authors proposed MRet (Matching for Retention), a dynamic learning-to-rank algorithm that learns personalized retention functions for each user. MRet then dynamically recommends users by maximizing the total expected retention gains for both the receiving and recommended users. The authors conducted numerical experiments on both synthetic and real-world data from a major online dating platform, demonstrating the efficacy of their method.

**Strengths:**

- I think the model studied in this paper is fairly novel and has practical relevance, as user retention is indeed aligned with real-world business metrics.
- The theoretical results appear sound, where the authors derive an nice relaxation of an NP-hard objective via concavity assumptions.
- The authors conducted numerical experiments on both synthetic and real-world data and provided very comprehensive analysis. In particular, the discussion provided nice insights into how fairness-centric methods differ from retention-centric approaches.

**Weaknesses:**

- The framework assumes access to the match probability $r(x,y)$, whereas in practice this must be estimated online. Although the paper acknowledges this limitation, it does not empirically assess robustness to estimation errors. Such errors could substantially degrade performance, and their impact deserves further investigation.
- The real-world validation is relatively limited, as the experiment uses a sampled subset of 1k × 1k users with imputed match probabilities rather than a fully deployed system. This setup demonstrates feasibility but does not provide evidence of live or large-scale performance.
- In this paper, fairness is presented primarily as a comparison to retention, but in practice I think it remains an important consideration for many platforms. Could fairness be incorporated alongside retention as a secondary objective within this framework?

**Questions:**

Related to the real-world experiments, I also have a conceptual question: on dating platforms, user retention and user satisfaction may not always align. For instance, if matching is highly effective, satisfied users may form lasting relationships and naturally stop using the platform. In such cases, is retention truly the right optimization objective? It would be valuable for the authors to discuss how platforms might balance retention with user success or satisfaction, and whether MRet can accommodate such trade-offs.

---

> ### Author Response · Authors · 2025-11-19
> **Official Comment by Authors**
>
> We appreciate the valuable and thoughtful feedback from the reviewer. We respond to the concrete questions and comments in detail below.
>
> > The framework assumes access to the match probability $r(x, y)$, whereas in practice this must be estimated online. Such errors could substantially degrade performance, and their impact deserves further investigation.
>
> We appreciate the interesting suggestion by the reviewer.
> **The problem of user retention would still exist even if the match probabilities are perfectly estimated**. This is why we focus solely on retention probabilities and not on match probabilities estimation.
>
> Nonetheless, as the reviewer mentions, match probability estimation is not perfect in reality, so we agree that it is important to test on noisy match probabilities. **We have added an experiment where the match probability $r(x, y)$ is perturbed with noise**.
> Specifically, we add noise $\varepsilon_{x,y}$ to the match probabilities to see the effect of match probability estimation error.
> $$\tilde r(x,y) = r(x,y) + \varepsilon_{x,y}, \qquad \varepsilon_{x,y} \sim \mathcal{U}[-\delta,,\delta],$$
> where ($\delta$) controls the noise magnitude. **The results summarized in the table below show that MRet yet retains the most users despite heavy noise.** We will add the results in the revised version.
>
> **Table: User retention rate on different math probability estimation noise**
> | method / 𝛿     | 0.0     | 0.1     | 0.2     | 0.3     | 0.4     | 0.5     |
> |-----------------|---------|---------|---------|---------|---------|---------|
> | Max Match       | 0.545   | 0.549   | 0.553   | 0.556   | 0.555   | 0.554   |
> | FairCo          | 0.585   | 0.582   | 0.581   | 0.577   | 0.578   | 0.581   |
> | Uniform         | 0.552   | 0.552   | 0.552   | 0.552   | 0.552   | 0.552   |
> | **MRet**        | **0.618** | **0.615** | **0.609** | **0.608** | **0.605** | **0.601** |
> | **MRet (best)** | **0.651**   | **0.646**   | **0.639**| **0.633**   | **0.628**   | **0.625**   |
>
>
> > The real-world validation is relatively limited, as the experiment uses a sampled subset of 1k × 1k users with imputed match probabilities rather than a fully deployed system.
>
> We appreciate the comment by the reviewer.
>
> First, to account for the reviewer’s concern, **we additionally conducted a real-world experiment on 5000 × 5000** users. As shown in the table below, MRet consistently outperforms all baselines in this larger setup. Crucially, the performance trends align perfectly with those in the 1000 × 1000 setting. This consistency confirms that our findings are robust to scale and that the dataset size used in the main text (which is similar to the scale of prior works [1, 2, 3, 4, 5]) serves as a representative proxy for larger interaction patterns. We will add the results in the revised version.
>
> **Table: User retention rate on 5000 x 5000 on different timesteps**
> | method               |  t= 1000  | t= 2000  | t= 3000  |
> |-----------------------|--------------|------------|-------------|
> | Uniform              | 0.751  |  0.607   | 0.507   |
> | Max Match         | 0.764  |  0.634   | 0.550  |
> | FairCo               | 0.764   |  0.629   | 0.538   |
> | **MRet**       |  **0.765** | **0.639** | **0.561** |
> | **MRet (best)**  | **0.770**  |  **0.649**   | **0.578**  |
>
> Regarding the reliance on imputation, we strictly adhered to the standard evaluation protocols established in recent reciprocal recommendation literature [1, 2, 3, 4, 5]. We adopted this widely accepted setting to ensure that our experimental methodology of leveraging imputed match probabilities is consistent with the standards of the field, thereby validating the reliability of our results within the community's established framework.
>
> [1] Su, Y., Bayoumi, M., Joachims, T. Optimizing Rankings for Recommendation in Matching Markets. WWW 2022.
>
> [2] Saini, R., Rusu, F., Johnston, T. PrivateJobMatch: A Privacy-Oriented Deferred Multi-Match Recommender System for Stable Employment. RecSys 2019.
>
> [3] Tomita, Y., Togashi, R., Hashizume, Y., Ohsaka, N. Fast and Examination-Agnostic Reciprocal Recommendation in Matching Markets. RecSys 2023.
>
> [4] Tomita, Y., Yokoyama, T. Fair Reciprocal Recommendation in Matching Markets. RecSys 2024.
>
> [5] Vitale, F., Parotsidis, N., Gentile, C. Online Reciprocal Recommendation with Theoretical Performance Guarantees. NeurIPS 2019.

---

> ### Author Response · Authors · 2025-11-19
> **Official Comment by Authors (cont'd)**
>
> > In this paper, fairness is presented primarily as a comparison to retention, but in practice I think it remains an important consideration for many platforms. Could fairness be incorporated alongside retention as a secondary objective within this framework?
>
> We appreciate the question from the reviewer. **Technically, fairness can be easily incorporated into MRet if needed for some reason.** As the reviewer has suggested, we can integrate fairness as a secondary objective by combining the MRet score with a fairness correction term as $$\text{Score}_\text{hybrid}(y) = \text{Score}(y) + \lambda \cdot \text{err}(y),$$where $\text{Score}(y)$ is the MRet scores (Eq. 13) and $\text{err}(y)$ is the fairness disparity (Eq. 6).
>
> **However, we would like to note that there is an inherent trade-off when considering fairness.** Enforcing fairness constraints inevitably restricts the optimization space, leading to a degradation in the primary objective of user retention. **Based on our experience collaborating with a real-world matching platform, platform sustainability depends primarily on user retention, as it is directly tied to revenue, whereas fairness considerations are not. Consequently, it is likely that extremely few platforms would sacrifice this critical metric to satisfy fairness, unless they are subject to strict external constraints like legal requirements.** Therefore, while MRet represents a flexible framework that can accommodate fairness if required, our paper focuses on retention as the unconstrained primary goal to reflect the reality of the platform.
>
> > on dating platforms, user retention and user satisfaction may not always align. For instance, if matching is highly effective, satisfied users may form lasting relationships and naturally stop using the platform. In such cases, is retention truly the right optimization objective? It would be valuable for the authors to discuss how platforms might balance retention with user success or satisfaction, and whether MRet can accommodate such trade-offs.
>
> We appreciate the insightful comment.
>
> **First, we emphasize that our framework is agnostic to the definition of the objective function.** MRet is a general framework that can optimize for user satisfaction simply by replacing the reward definition (the function $f$ in Eq. 9) with a satisfaction metric, without changing the underlying algorithm.
>
> **We focused on optimizing retention in this paper primarily due to its direct alignment with platform sustainability.** For most subscription-based platforms, retention is the primary driver of revenue. As the reviewer correctly pointed out, "satisfaction" (e.g., finding a permanent partner) often leads to immediate churn, which can be misaligned with the platform's business incentives. Furthermore, retention signals are relatively easy to collect through observed user behavior logs, whereas satisfaction is subjective and much harder to obtain in dense form because it requires explicit feedback, making it more difficult to optimize.
>
> **In summary, while MRet is fully capable of accommodating satisfaction as the objective if it is desired and observed as part of the logged data, we constructed our formulation and experiments around retention due to its practical applicability and observability in real-world settings.** We will clarify this point in the revision.

---

> ### Author Response · Authors · 2025-11-27
> **We would appreciate knowing whether our responses have sufficiently addressed the reviewer’s initial concerns.**
>
> We greatly appreciate the thoughtful review by the reviewer once again.
>
> For the reviewer’s information, we have updated the draft and included all the additional experiments suggested by the reviewer and other reviewers. The changes are written in red. More specifically:
>
> * We empirically demonstrated that **MRet remains the best solution even when there is noise $\delta$ in the match probabilities.**
>
> * We added experimental results with a **larger sample size 5000 x 5000** in the appendix.
>
> * We moved the Related Works section to the main text, with some more references to the related literature.
>
> * In the experiments section, we added explained why [Tomita and Yokoyama, 2024] cannot be compared as a baseline.
>
> * We added experimental results with weighted retention objectives $w(x)$ and $w(y)$ in the appendix.
>
> * Reworded from "popularity bias" to "popularity skew" to prevent confusion.
>
> **In our responses, we believe we have addressed all concerns raised in the initial review. At this point, we are not aware of any remaining issues that would prevent an updated assessment. If there are still concerns that motivate the current score of 4, we would greatly appreciate the opportunity to understand them more fully, which we believe would be highly valuable for further improving the paper.** We would be grateful if the reviewer could clarify any remaining issues soon, especially as the discussion deadline is approaching.

---

### Official Review · Reviewer_M3j6 · 2025-10-28

**Soundness:** 2
**Presentation:** 3
**Contribution:** 3
**Rating:** 6
**Confidence:** 3

**Summary:**

This paper departs from the classic goals of two-sided matching (e.g., stable matchings, utility or welfare maximization, fairness constraints) by proposing a new objective: user‐retention (or platform‐driven “stickiness”) in a two-sided matchmaking setting (e.g., online dating, ride-sharing, job matching). The authors argue that match count or immediate welfare may not correlate strongly with long‐term retention, and instead propose modeling a personalized retention curve for each match pair (or each agent × partner type) that estimates the probability of continued engagement over time. They then formulate a matching/allocating algorithm, MRet, which assigns scarce match opportunities not simply to maximize immediate matching or fairness but to maximize expected retention gain subject to capacity constraints. Empirically, they validate the retention‐curve learning (from observational data) and test MRet on a large real‐world dataset (from a dating-platform) plus synthetic simulations. The results suggest that MRet can increase overall retention by focusing matches where retention impact is large, and still provide acceptable fairness outcomes (e.g., representation across groups).

**Strengths:**

1-  Many matching/recommender systems (dating, jobs, rides, tutors) care about retention, not just initial matching. Framing retention‐maximization in two‐sided matching is novel in the ML/optimization space.

2- Learning per‐agent or per‐pair retention probability over time is a creative step that bridges predictive modeling with matching allocation.

3- The shift from maximizing number of matches/fairness to maximizing expected retention under capacity constraints is clean and actionable.

4- Real‐world data from a large-scale platform, showing retention lift and reasonably good fairness outcomes, is a strong edge for an applied paper.

5- The authors do not ignore fairness; they include group‐level analyses and discuss retention trade-offs across demographic slices.

**Weaknesses:**

1- The paper lacks formal results on the complexity of the retention‐objective matching problem, or approximation bounds/algorithms for large‐scale graphs. If the objective is non-linear or non-additive, this becomes non-trivial. The submission would be stronger if it clearly stated whether the problem is polynomially solvable (e.g., weighted bipartite matching) or NP‐hard and if so, provided a provably efficient approximation.

2- Learning retention curves is central. However, the paper provides limited diagnostics of how accurate those curves are (prediction error, cross‐validation, time‐shift evaluation). Also, the domain dependence (single platform dataset) raises questions: how does the method generalize to new user cohorts, or across platforms?

3- While fairness is discussed, the core algorithm (MRet) optimizes retention. If certain groups inherently have lower retention predictions, the algorithm may preferentially allocate to higher‐retention groups, exacerbating representation/fairness issues. More explicit trade‐off analysis (worst‐group retention, disparity curves) would strengthen the fairness dimension.

4- The evaluation would benefit from stronger comparative baselines beyond standard matching: for example, match‐volume maximization with fairness constraints, heuristic retention prioritization without modeling, and analyses of small vs. large capacity constraints. Also, ablation studies (e.g., retention modeling on/off, fairness constraints on/off) would help attribute gains.

5- It is unclear how the approach scales to many‐to‐many matching or large quotas, or domains where retention is less measurable or delayed. Some simulation/transfer experiments would help.

**Questions:**

1- Is the allocation problem solved as a linear assignment with weights = expected retention gains, or is it more complex (e.g., non-additive, dependent pairs)? If the latter, what solver is used, and what is the worst‐case runtime?

2- What features are used to predict retention? How was the model validated (train/dev/test splits, cross‐validation, hold‐out platform cohort)? What is the typical prediction error (e.g., RMSE, calibration by bins)?

3- How sensitive are results to the capacity constraint (e.g., limited matches per period)? Did you vary the quota regime and show retention gains scale?

4- Given that retention predictions may correlate with demographic/time-on‐platform features, how do you ensure that optimizing retention doesn’t disproportionately disadvantage historically marginalised groups? Could you offer fairness constraints (min‐quota, parity) and show corresponding retention/representation curves?

5- If user cohorts shift or retention distributions change over time (e.g., seasonality, platform evolution), how robust is the learned retention model? Did you evaluate transfer across time windows or hold‐back users?

6- Have you applied MRet beyond the dating platform domain (e.g., job-matching, ride‐sharing) or simulated different domains to test generality?

7- Did you compare to simpler heuristics (e.g., match first, then sort by predicted retention gain greedily) rather than full model + allocation? And what is the incremental benefit of the full pipeline?

---

> ### Author Response · Authors · 2025-11-19
> **Official Comment by Authors**
>
> We appreciate the valuable and thoughtful feedback from the reviewer. We respond to the concrete questions and comments in detail below.
>
> > The submission would be stronger if it clearly stated whether the problem is polynomially solvable (e.g., weighted bipartite matching) or NP‐hard and if so, provided a provably efficient approximation.
>
> > Is the allocation problem solved as a linear assignment with weights = expected retention gains, or is it more complex (e.g., non-additive, dependent pairs)? If the latter, what solver is used, and what is the worst‐case runtime?
>
> We appreciate the opportunity to clarify the theoretical complexity and efficiency of our framework.
>
> **The optimization problem is indeed NP-hard**, as we state in the main text. This is because the retention gain from a specific match depends on the user's current cumulative match count, creating complex dependencies that prevent simple linear assignment solutions.
>
> **To address this, MRet is designed as a provably efficient and effective approximation**. By deriving a theoretical lower bound of the objective using Jensen's inequality (Lemma 1 & 2) , we transform the complex combinatorial problem into a sorting problem. **Since MRet takes the form of argsort, the worst-case runtime is $O(N \log N)$**, which is scalable for real-time applications.
>
> > 2- What features are used to predict retention? How was the model validated (train/dev/test splits, cross‐validation, hold‐out platform cohort)? What is the typical prediction error (e.g., RMSE, calibration by bins)?
>
> We are happy to clarify. To predict retention, we used user vectors (drawn from a standard normal distribution for synthetic data and derived from profile attributes for real-world data) and match counts as input features. Regarding validation, we trained the model using data sampled from simulations and evaluated its performance by deploying the trained model in simulation runs. Additionally, given that this is a binary classification task, we utilized Binary Cross-Entropy as the loss function.
>
> > 3- How sensitive are results to the capacity constraint (e.g., limited matches per period)? Did you vary the quota regime and show retention gains scale?
>
> We are happy to address the concern.
> In Figure 5 in Appendix E, we show results on varying numbers of users. **This experiment explicitly changes the number of matches given to a user. From the results, we observe that MRet is robust to such capacity constraints.**
>
> > 3- While fairness is discussed, the core algorithm (MRet) optimizes retention. If certain groups inherently have lower retention predictions, the algorithm may preferentially allocate to higher‐retention groups, exacerbating representation/fairness issues. More explicit trade‐off analysis (worst‐group retention, disparity curves) would strengthen the fairness dimension.
>
> > 4- Given that retention predictions may correlate with demographic/time-on‐platform features, how do you ensure that optimizing retention doesn’t disproportionately disadvantage historically marginalised groups? Could you offer fairness constraints (min‐quota, parity) and show corresponding retention/representation curves?
>
> We appreciate the question and remark. We acknowledge that unconstrained retention optimization may inadvertently favor high-retention groups. However, **our primary contribution in this work is to establish the theoretical and empirical framework for retention maximization itself, filling a fundamental gap in the literature.**
>
> **We view fairness as an orthogonal objective that represents a different research dimension.** As discussed in our response to Reviewer 4, MRet is flexible enough to incorporate fairness constraints if specific demographic parity is required. We decided to focus on the unconstrained setting to clearly demonstrate the effectiveness of the core retention mechanism, leaving the complex analysis of fairness-retention trade-offs for future work dedicated to that specific topic.

---

> ### Author Response · Authors · 2025-11-19
> **Official Comment by Authors (cont'd)**
>
> > 5- If user cohorts shift or retention distributions change over time (e.g., seasonality, platform evolution), how robust is the learned retention model? Did you evaluate transfer across time windows or hold‐back users?
>
> This is a great question.
> **MRet is inherently designed to handle temporal dynamics.** Unlike static recommendation models that might overfit to a specific time window, MRet is an _online planning algorithm_. It dynamically evaluates the marginal retention gain at each time step $\tau$ based on the current state of accumulated matches (as defined in Eq. 9). **This mechanism allows the algorithm to naturally adapt its ranking strategy in real-time, even as user cohorts shift or interaction patterns evolve, without requiring constant retraining of the underlying retention function.**
>
> **To empirically verify this robustness against shifts, we conducted an experiment introducing a time-dependent popularity bias, which simulates a severe form of distribution shift.** In this experiment, we introduce a time-dependent popularity bias, defining the match probability at time step $t$ as
> $$r_t(x, y) = (1 - \kappa)r_{\text{base}}(x, y) + \kappa pop_t(x)pop_t(y).$$
> We assign each user $x \in \mathcal{X}$ and $y \in \mathcal{Y}$ one of three popularity trajectories: constant, increasing, or decreasing. For each user $i$, we sample an initial popularity $c_i \sim \mathrm{Uniform}(0,1)$ and set a fixed slope of $0.5$.Letting $\tau_t \in [0,1]$ denote the normalized time step index, we define the time-varying popularity as
> $$pop_t(i) =
> \begin{cases}
> c_i & \text{constant}, \\\\
> c_i + 0.5 \tau_t & \text{increasing}, \\\\
> c_i - 0.5 \tau_t & \text{decreasing}.
> \end{cases}$$
> We then clip the resulting $pop_t(i)$ to the range $[0, 1]$ to ensure it remains within a valid bound. **As the results in the table below show, and MRet outperforms all baselines, demonstrating its effective adaptation to popularity drifts.** We will add this in the revision.
>
> **Table: User retention rate on dynamic popularity change on different timesteps**
> | method               | t= 500  | t= 1000 | t= 1500  | t= 2000  |
> |-----------------------|-----------|------------|-------------|-------------|
> | Max Match        |  0.692  |  0.570   | 0.509  | 0.475  |
> | FairCo              | 0.697  |  0.586   | 0.530  | 0.499   |
> | Uniform             |  0.685   |  0.557   | 0.496  | 0.462   |
> | **MRet**         | **0.699**  | **0.591** | **0.545**  |**0.521** |
> | **MRet (best)** |  **0.710**  |  **0.608**   | **0.564**  | **0.546**  |
>
> > 6- Have you applied MRet beyond the dating platform domain (e.g., job-matching, ride‐sharing) or simulated different domains to test generality?
>
> We are happy to answer the reviewer’s question. **In our synthetic simulation experiment, we did not focus on a particular domain.** The setting is a general setting adaptable to any of the domains in two-sided matching. We are happy to do a simulation experiment if the reviewer could specify how the setting of a particular domain (e.g., job-matching, ride‐sharing, online dating) _meaningfully_ differs from the setting we synthesize.
>
> > 7- Did you compare to simpler heuristics (e.g., match first, then sort by predicted retention gain greedily) rather than full model + allocation? And what is the incremental benefit of the full pipeline?
>
> We may be confused by what the reviewer means by a simpler heuristic.
> **MRet does exactly what the reviewer proposes: match first, then sort by predicted retention gain greedily.** MRet is a simple, effective, yet theoretically grounded method to address the problem.

---

### Official Review · Reviewer_ioZz · 2025-10-29

**Soundness:** 2
**Presentation:** 2
**Contribution:** 2
**Rating:** 4
**Confidence:** 3

**Summary:**

This paper reframes online two-sided matching around user retention rather than match count or fairness. Under the assumption that per-user retention as a function of cumulative matches is monotone and concave, the authors apply a Jensen/linear lower bound to derive an additive, argsortable score that accounts for both receiver- and candidate-side marginal retention gains. Experiments on synthetic data and a small real dataset report retention improvements.

**Strengths:**

- Elevates long-term user retention over proxy objectives like match count or exposure parity.
- An interesting setting that explicitly models both receiver and candidate utilities to enhance long-term retention.
- Clean derivation and standard components make the method easy to code and to reproduce end-to-end.

**Weaknesses:**

1. **Motivation/positioning is unclear. Fairness appears as a “solution” without a precise link to retention.**

The paper treats fairness methods as peers to retention optimization, but never clarifies the key retention concepts (e.g., “satisfaction thresholds,” “marginal retention slope,” “early low-match churn”) nor how these relate to fairness (guardrail vs competing target).

2. **Related work is late and under-covered, blurring the contribution boundary.**

Retention optimization and reciprocal-fairness are direct neighbors; coverage is thin and appears too late. The following are some examples:
> - Wang Y, Sun P, Ma W, et al. Intersectional two-sided fairness in recommendation[C]//Proceedings of the ACM Web Conference 2024. 2024: 3609-3620.
> - Biswas A, Patro G K, Ganguly N, et al. Toward fair recommendation in two-sided platforms[J]. ACM Transactions on the Web (TWEB), 2021, 16(2): 1-34.

3. **Target misaligned. Examples/experiments focus on popularity bias, but the primary comparator optimizes disparity**

Most setups reveal a ‘rich-get-richer’ dynamic, yet the fairness baseline optimizes exposure disparity rather than marginal retention for low-match users and does not directly address popularity bias. This misalignment structurally favors MRet.

4. **Baselines are limited and their selection lacks rationale.**

Several cited works are not used as baselines; Max Match/FairCo’s origin/implementation/tuning is under-specified.

5. **Real-world setting is narrow and assumption-heavy**

Small sample sizes(1000 x 1000), heavy reliance on missing-value imputation, coarse retention labels (e.g., next-month login), and no off-policy or online evidence.

**Questions:**

1. In dating, a “successful match” often triggers intentional departure. Did you distinguish success-exit from disengagement churn?
2. Do you model popularity as time-varying $p_u(t)$ (recency, seasonality, inactivity decay)? Please provide dynamics of popularity over time and test whether your method adapts when popularity drifts.

---

> ### Author Response · Authors · 2025-11-19
> **Official Comment by Authors**
>
> We appreciate the valuable and thoughtful feedback from the reviewer. We respond to the concrete questions and comments in detail below.
>
> > Motivation/positioning is unclear. Fairness appears as a “solution” without a precise link to retention.
>
> We would like to address a potential critical misunderstanding by the reviewer. **“Fairness does not have a link to retention” is exactly what we are claiming in this paper.**
>
> The issue of fairness axioms is that maximization of fairness does not have a positive effect in itself, so we claim that this is not a good _objective_. Then, the next question is whether fairness is a good _solution_ to maximize objectives that are actually beneficial, like user retention. **We argue, through experiments, that fairness is not a good solution for user retention either, exactly because it has no precise link to retention and leaves the optimization of retention up to chance.**
> While fairness is often naively used in practice, this paper warns that fairness can often be neither a good objective nor a good solution.
>
> > Target misaligned. Examples/experiments focus on popularity bias, but the primary comparator optimizes disparity.
> > Most setups reveal a ‘rich-get-richer’ dynamic, yet the fairness baseline optimizes exposure disparity rather than marginal retention for low-match users and does not directly address popularity bias. This misalignment structurally favors MRet.
>
> We respectfully disagree with the reviewer on this point.
>
> First, **our experimental setup does not simulate a “rich-get-richer” dynamic**. In our experiments, the match probability $r(x,y)$ is static and is not updated. Therefore, **the "popularity bias" in our experiments refers to the inherent skew in user utility distributions (static inequality), not a rich-get-richer feedback loop where exposure changes future relevance**. Since FairCo is specifically designed to mitigate exposure disparity, it is a valid and standard baseline for handling such skewed utility distributions. Furthermore, contrary to the reviewer's implication that the baseline fails to address bias, FairCo _is_ precisely built to account for the rich-get-richer problem, as stated in their paper.
> “In this paper, we present the first dynamic LTR algorithm – called FairCo – that overcomes rich-get-richer dynamics” [Morik et al., 2020, paragraph 3 Introduction].
>
> Second, regarding the remark that our experiments "focus on popularity bias," we emphasize that this bias is introduced as an environmental setting to simulate realistic platform conditions, not as the optimization target. The reviewer implies that the goal should be to directly address this bias, but our fundamental argument is that **addressing exposure disparity is structurally misaligned with the platform objective**.
>
> Most importantly, as we have also discussed in the previous point, **the target misalignment by fairness algorithms (and dealing with popularity bias) is exactly what we want to point out in this paper**. We would like to clarify that MRet outperforming the other “misaligned” baselines is therefore _natural_ and _expected_ (e.g., we state this explicitly in lines 29 and 94). Again, a major contribution of the paper is to inform and warn the readers that naively applying fairness algorithms could create a crucial misalignment with the platform objective.
>
> [Morik et al., 2020] Morik M, Singh A, Hong J, et al. Controlling fairness and bias in dynamic learning-to-rank. Proceedings of the 43rd International ACM SIGIR 2020: 429–438.
>
> > Baselines are limited and their selection lacks rationale.
>
> We appreciate the remark by the reviewer. However, in terms of user retention, which is the platform’s goal, **all fairness algorithms are misaligned and are not rational, including FairCo**. Since fairness is misaligned with the objective, it will leave the optimization of retention up to chance. Therefore, MRet outperforming the other “misaligned” baselines is completely natural and expected. The comparison with FairCo confirms this, and we choose FairCo simply because it is the most widely used fair dynamic LTR algorithm, to the best of our knowledge. The comparison against the uniform (random) baseline and the Max Match baseline confirms that the fairness baseline may still be better than not handling the problem.
>
> From the standpoint that all fairness algorithms are misaligned, we would like to humbly ask what fairness baseline would be a _rational_. If the reviewer has more specific additional baselines to produce more advantageous and insightful experiments for the specific problem of reciprocal retention maximization in dynamic LTR compared to what we have provided, we would need to understand those designs in detail to have a more constructive discussion. If the reviewer cannot specify concrete additional baselines and articulate the reasons for including them, we are left with no choice but to conclude that our current baseline selection is sufficient.

---

> ### Author Response · Authors · 2025-11-19
> **Official Comment by Authors (cont'd)**
>
> > Related work is late and under-covered, blurring the contribution boundary.
>
> Thank you for providing these valuable references. We will cite them in the revised version of the paper. **We would like to clarify, however, that the papers suggested by the reviewer [1, 2] address two-sided fairness that does not consider mutual preferences, which differs from our setting of reciprocal recommendations.** Due to this important distinction, we believe that fully covering the broader literature on _non-reciprocal_ two-sided recommendation may not directly serve the focus of our work.
>
> That said, we fully acknowledge the conceptual relevance of this line of research, and this is why we already cite several representative works from this area, such as [3, 4], in the main text. We hope this clarifies how our positioning relates to the existing literature.
>
> [1] Wang Y, Sun P, Ma W, et al. Intersectional two-sided fairness in recommendation[C]//Proceedings of the ACM Web Conference 2024. 2024: 3609-3620.
>
> [2] Biswas A, Patro G K, Ganguly N, et al. Toward fair recommendation in two-sided platforms[J]. ACM Transactions on the Web (TWEB), 2021, 16(2): 1-34.
>
> [3] Devic S, Kempe D, Sharan V, et al. Fairness in matching under uncertainty[C]//Proceedings of the 40th International Conference on Machine Learning (ICML 2023). 2023: 7775-7794.
>
> [4] Do V, Corbett-Davies S, Atif J, et al. Two-sided fairness in rankings via Lorenz dominance[C]//Advances in Neural Information Processing Systems 34 (NeurIPS 2021). 2021: 8596-8608.
>
> > Real-world setting is narrow and assumption-heavy:
> Small sample sizes(1000 x 1000), heavy reliance on missing-value imputation, coarse retention labels (e.g., next-month login), and no off-policy or online evidence.
>
> We appreciate the remarks by the reviewer.
> First, to address the reviewer’s concern on sample sizes, **we conducted real-world experiments on 5000 x 5000 below**. We observe that MRet consistently outperforms all baselines in this larger setup. Crucially, the performance trends align with those in the 1000 x 1000 setting. This consistency confirms that our findings are robust to scale and that the smaller dataset used in the main text (which aligns with the scale of prior works [5, 6, 7, 8, 9]) serves as a representative proxy for larger interaction patterns. We will add this in the revised version
>
> **Table: User retention rate on 5000 x 5000 on different timesteps**
> | method               |  t= 1000  | t= 2000  | t= 3000  |
> |-----------------------|--------------|------------|-------------|
> | Uniform              | 0.751  |  0.607   | 0.507   |
> | Max Match         | 0.764  |  0.634   | 0.550  |
> | FairCo               | 0.764   |  0.629   | 0.538   |
> | **MRet**       |  **0.765** | **0.639** | **0.561** |
> | **MRet (best)**  | **0.770** |  **0.649**   | **0.578**  |
>
> Second, regarding the reliance on imputation and the use of offline evaluation, we strictly adhered to the standard evaluation protocols established in the reciprocal recommendation literature [5, 6, 7, 8, 9]. We adopted this widely accepted setting to ensure that our experimental methodology is consistent with the standards of the field, thereby validating the reliability of our results within the community's established framework.
>
> Finally, **the sparsity of the retention label is inherently case-dependent and therefore difficult to discuss in a general manner here. What is important for our work is that our method is flexible and can accommodate any definition of retention**. In other words, one may choose a shorter-term retention metric that is easier to optimize, or a longer-term metric that tends to be sparser and more challenging to optimize, and our approach can adapt to either. By contrast, baselines such as fairness methods or Max Match do not directly optimize retention, making such flexible switching of retention definitions fundamentally infeasible for them.
>
> [5] Su, Y., Bayoumi, M., Joachims, T. Optimizing Rankings for Recommendation in Matching Markets. WWW 2022.
>
> [6] Saini, R., Rusu, F., Johnston, T. PrivateJobMatch: A Privacy-Oriented Deferred Multi-Match Recommender System for Stable Employment. RecSys 2019.
>
> [7] Tomita, Y., Togashi, R., Hashizume, Y., Ohsaka, N. Fast and Examination-Agnostic Reciprocal Recommendation in Matching Markets. RecSys 2023.
>
> [8] Tomita, Y., Yokoyama, T. Fair Reciprocal Recommendation in Matching Markets. RecSys 2024 (to appear / arXiv:2409.00720).
>
> [9] Vitale, F., Parotsidis, N., Gentile, C. Online Reciprocal Recommendation with Theoretical Performance Guarantees. NeurIPS 2019.

---

> > ### Author Response · Authors · 2025-11-19
> > **Official Comment by Authors (cont'd)**
> >
> > > In dating, a “successful match” often triggers intentional departure. Did you distinguish success-exit from disengagement churn?
> >
> > This is a great question.
> > We could not distinguish success-exit from disengagement churn without explicit feedback. Nonetheless, due to the nature of the regression model that models the retention probability (along with the concavity function), **MRet is naturally able to model the success-exit too if it has a large effect**.
> >
> > However, **if the platform’s priority is to have more success-exits, user satisfaction _can_ be treated as an objective in MRet**. This can be done simply by replacing the reward signals from user retention to user satisfaction. Please note, however, that this would only be possible if the platform can observe enough explicit feedback from the users. Either way, MRet does not require any change to its algorithm.
> >
> > > Do you model popularity as time-varying $p_u(t)$ (recency, seasonality, inactivity decay)? Please provide dynamics of popularity over time and test whether your method adapts when popularity drifts.
> >
> > We appreciate the constructive suggestion. **This experiment provides a valuable opportunity to highlight a key strength of MRet, which is its adaptability.**
> >
> > **While our main text focuses on validating the effectiveness of the retention objective in standard settings, MRet is naturally designed to handle dynamics by optimizing the marginal retention gain at each time step $\tau$ (Eq. 9).**
> >
> > We conducted an additional experiment to test how MRet adapts when popularity drifts over time. In this experiment, we introduce a time-dependent popularity bias, defining the match probability at time step $t$ as
> > $$r_t(x, y) = (1 - \kappa) r_{\text{base}}(x, y) + \kappa pop_t(x) pop_t(y).$$
> > We assign each user $x \in \mathcal{X}$ and $y \in \mathcal{Y}$ one of three popularity trajectories: constant, increasing, or decreasing. For each user $i$, we sample an initial popularity $c_i \sim \mathrm{Uniform}(0,1)$ and set a fixed slope of $0.5$.Letting $\tau_t \in [0,1]$ denote the normalized time step index, we define the time-varying popularity as $$pop_t(i) =
> > \begin{cases}
> > c_i & \text{constant}, \\\\
> > c_i + 0.5 \tau_t & \text{increasing}, \\\\
> > c_i - 0.5 \tau_t & \text{decreasing}.
> > \end{cases}$$We then clip the resulting $pop_t(i)$ to the range $[0, 1]$ to ensure it remains within a valid bound. As the results in the table below show, and MRet outperforms all baselines in terms of user retention rate, demonstrating its effective adaptation to popularity drifts. **This confirms that MRet successfully adapts to popularity drifts, as expected from its dynamic design.** We will add this in the revision.
> >
> > **Table: User retention rate on dynamic popularity change**
> > | method               | t= 500  | t= 1000 | t= 1500  | t= 2000  |
> > |-----------------------|-----------|------------|-------------|-------------|
> > | Max Match        |  0.692  |  0.570   | 0.509  | 0.475  |
> > | FairCo              | 0.697  |  0.586   | 0.530  | 0.499   |
> > | Uniform             |  0.685   |  0.557   | 0.496  | 0.462   |
> > | **MRet**         | **0.699**  | **0.591** | **0.545**  |**0.521** |
> > | **MRet (best)** |  **0.710**  |  **0.608**   | **0.564**  | **0.546**  |

---

> ### Comment · Reviewer_ioZz · 2025-11-22
> **Response to authors**
>
> I thank the authors for their detailed response. I am pleased to see that **W5, Q1, and Q2 have been addressed.**
>
> However, I remain unconvinced regarding **W1–W4**. I want to emphasize that I find the proposed method itself to be reasonable and technically sound. My concerns are strictly regarding the **motivation** and **experimental evaluation.** It seems my original points may have been misunderstood, so I would like to restate them clearly to help improve the paper's positioning and rigor.
>
> ## Regarding W1 (Motivation & Storytelling)
> The authors state in the rebuttal that *"Fairness does not have a link to retention"* is exactly what we are claiming. The paper also states: *"In real-world practice... casually choosing fairness formulations... leaves the optimization of retention up to luck."*
>
> My concern lies precisely here. I fully agree that fairness is a misaligned objective for retention. However, proving that a misaligned objective performs poorly does not inherently strengthen the contribution of MRet. Unless using fairness metrics to solve retention problems is common practice in Reciprocal Recommendation literature, this comparison feels like a "strawman" argument.
>
> - **The Issue:** The current storytelling implies the paper’s contribution partially stems from outperforming fairness methods on retention, despite these methods not being designed for that objective.
>
> - **The Reality:** The core contribution is the reciprocal utility optimization mechanism itself. Framing the entire motivation around "beating fairness" feels like a detour.
>
> - **Suggestion:** The paper would be much stronger if it focused on how MRet solves retention better than standard Reciprocal Recommendation methods (which are the true status quo) or general retention-oriented works, rather than focusing so heavily on the misalignment of fairness.
>
> ## Regarding W2 (Experimental Design & Bias):
> I have deep reservations about the mechanism behind the FairCo comparison in the synthetic experiments.
> In the experimental setup, the relevance score $r(x,y)$ is explicitly constructed to contain **popularity bias** (as stated in Section 4.1: "The second term introduces a popularity bias... $r_{pop}(x,y) = pop_x \cdot pop_y$").
>
> My concern stems from how FairCo interprets this score. Typically, fairness-of-exposure methods like FairCo treat the provided relevance score (here, $r(x,y)$) as the "merit" or "utility" that exposure should be proportional to.
> - **The Speculation:** If FairCo is indeed optimizing for exposure proportional to this $r(x,y)$ as its objective, it would systematically allocate more exposure to popular users, simply because their $r$ values are artificially higher due to the injected bias.
> - **The Implication:** This mechanism would explain the paper's observation that FairCo *"allocates more matches to already popular users"*. It suggests that the baseline's poor performance on retention might not be due to the method's failure, but rather because it is successfully optimizing a metric that—in this specific simulation—reinforces popularity bias.
>
> This raises the question of whether comparing against a fairness method optimizing this specific $r(x,y)$ provides a meaningful benchmark for retention, as the popularity bias seems to be encoded in the input metric itself.
>
> ## Regarding W3 (Baseline Rationale)
>
> I do not argue that the chosen baselines are "wrong," but the rationale for choosing them must be explicit in the main text, not just the rebuttal.
>
> Moreover, the authors state they chose FairCo as it is the "most widely used." However, FairCo (2020) is a general LTR method. The authors cited "Fair Reciprocal Recommendation in Matching Markets" (Tomita & Yokoyama, 2024) in the related work. This 2024 work is **more recent, fully aligned with the reciprocal recommendation topic, and shares a very similar setting (dating/synthetic+online).**
>
> Even if fairness is misaligned, why only compare against a 2020 general LTR method without comparing a 2024 domain-specific State-of-the-Art method? If the 2024 method is unsuitable, the paper should explicitly discuss why in the experiment details.
>
> - Note: The citation for FairCo is indeed missing in the experiment section (Page 7, line 347) .
>
> Finally, as mentioned in W1, the lack of standard Reciprocal Recommendation baselines (beyond heuristic Uniform/MaxMatch) and optional adaptations of general long-term engagement / retention-oriented methods remains a gap.
>
> ## Regarding W4 (Related Work Placement)
> I firmly believe that placing Related Work in the Appendix is not a standard practice, as it prevents readers from clearly understanding the paper's scope and positioning within the literature. Given that the rebuttal revision allows for 10 pages, **I strongly suggest moving the Related Work section back to the main body.**
>
> **I hope these clarifications help the authors understand that my critique aims to strengthen the paper's narrative and empirical validity.**

---

> ### Author Response · Authors · 2025-11-24
> **Official Comment by Authors**
>
> We appreciate the response from the reviewer, and there are many interesting points to discuss in the responses. We will address them below.
>
>
> > Regarding W1
>
> > Proving that a misaligned objective performs poorly does not inherently strengthen the contribution of MRet. Unless using fairness metrics to solve retention problems is common practice in Reciprocal Recommendation literature, this comparison feels like a "strawman" argument.
>
>
> > The Issue: The current storytelling implies the paper’s contribution partially stems from outperforming fairness methods on retention, despite these methods not being designed for that objective.
>
>
> > The Reality: The core contribution is the reciprocal utility optimization mechanism itself. Framing the entire motivation around "beating fairness" feels like a detour.
>
>
> > Suggestion: The paper would be much stronger if it focused on how MRet solves retention better than standard Reciprocal Recommendation methods (which are the true status quo) or general retention-oriented works, rather than focusing so heavily on the misalignment of fairness.
>
> We appreciate the clarification by the reviewer.
>
> **We would like to clarify that we do not place our contribution to "beating fairness."** As we state in the main text, **MRet outperforming the baselines is completely expected and natural**. We do not claim our contribution here.
>
> We would like to reclarify the position of our method to address the reviewer’s concern.
>
> Most studies in the reciprocal recommendation settings seek to maximize the number of matches, and **there are no methods that aim to maximize retention**. The fairness methods are the only ones with a close motivation (and seemingly effective), as they _indirectly_ consider user satisfaction. **Thus, our main contribution is to define the new problem, provide an objective function, and to propose an algorithm that is strong both theoretically and empirically.**
>
> Now, we are not aware of any “general retention-oriented works” either that is clearly applicable. If the reviewer has a specific baseline in mind, along with guidance on how it could be integrated into our setting, we would greatly appreciate learning more about it so that we can accurately address the reviewer’s concern. Please do note that the reciprocal setting makes the optimization problem **_very challenging_** for any of the “general retention-oriented works” to be applicable, because of the NP-hardness (Eq. 9).

---

> ### Author Response · Authors · 2025-11-24
> **Official Comment by Authors (cont'd)**
>
> > Regarding popularity bias (W2)
>
> We answer this point separately from the rest of W2, since it seems like there was a possible mismatch in the understanding by the both of us in the terminology “popularity bias.”
>
> The common meaning of popularity bias (which the reviewer is referring to) is where “popular users receive more exposure than their underlying attractiveness would justify.”
>
> The popularity bias that we introduced in the synthetic experiment was simply to set a realistic setting where “some people are more popular than others.” As the reviewer has pointed out, this may have been very confusing. **We appreciate the remark and will modify the name of it to “popularity skew” in the revision.**
>
> > Regarding W2:
>
> > My concern stems from how FairCo interprets this score. Typically, fairness-of-exposure methods like FairCo treat the provided relevance score (here, ) as the "merit" or "utility" that exposure should be proportional to.
>
>
> > The Speculation: If FairCo is indeed optimizing for exposure proportional to this  as its objective, it would systematically allocate more exposure to popular users, simply because their  values are artificially higher due to the injected bias.
>
>
> > The Implication: This mechanism would explain the paper's observation that FairCo "allocates more matches to already popular users". It suggests that the baseline's poor performance on retention might not be due to the method's failure, but rather because it is successfully optimizing a metric that—in this specific simulation—reinforces popularity bias.
>
>
> > This raises the question of whether comparing against a fairness method optimizing this specific provides a meaningful benchmark for retention, as the popularity bias seems to be encoded in the input metric itself.
>
> We respectfully disagree with the reviewer’s point that FairCo reinforces popularity bias.
>
> **FairCo _is_ explicitly an algorithm that minimizes popularity bias.** The reviewer is right to notice that FairCo allocates more to popular users and less to unpopular users, because this is what popularity bias is all about: “popular users receive more exposure than their underlying attractiveness would justify.” In other words, a perfectly non-popularity-biased is a condition where “users gain justifiable ” The objective of FairCo is the exact illustration of a perfectly non-popularity-biased ranking,
>
>
>
> $$\frac{\frac{1}{\tau}\sum_{t=1}^{\tau} \mathrm{Exp}_{t} (y_i)}{\mathbb{E} _{p(x)}[r(x, y_i)]} =\frac{ \frac{1}{\tau} \sum _{t=1}^{\tau}  \mathrm{Exp} _{t} (y_j) }{\mathbb{E} _{p(x)}[r(x, y_j)]}.$$
>
>
> **FairCo is introduced because too much exposure is given to popular users if we rank naively by relevance. This is not to place more allocations to the popular users, but to allocate less to them.**
>
> Also, we do test a situation where the popularity skew is $\kappa=0$ in Figure 4. In this setting, all users have equally popular. Moreover, we also observe that testing out different “fairness-level” hyperparameter does not change our conclusion (Figure 8, Appendix E). Even in these setting, MRet outperformed the baselines.

---

> ### Author Response · Authors · 2025-11-24
> **Official Comment by Authors (cont'd)**
>
> > Even if fairness is misaligned, why compare against a 2020 general LTR method instead of a 2024 domain-specific State-of-the-Art method? If the 2024 method is unsuitable, the paper should explicitly discuss why in the experiment details.
>
> We appreciate the question and suggestion by the reviewer, and we are happy to clarify.
>
> [Tomita et al., 2024] is strictly not comparable because of the difference in settings. Specifically, our setting is a Dynamic LTR setting, where we do not know which user visits the platform at what time. **In contrast, [Tomita et al., 2024] (and the spectrum of the prior reciprocal matching studies) calculate a doubly stochastic matrix.** [Tomita et al., 2024] do not care about who arrives at what time (i.e., the batch recommendation problem rather than dynamic). Calculating this doubly stochastic matrix everytime is not realistic at all, since the calculation takes exponential time. We hope this clarifies the reason why the comparison is not done. **While we do note about this in the related works (lines 687~690), but for clarity, we will also explain this in the experiment section.** We appreciate the suggestion.
>
> > Finally, as mentioned in W1, the lack of standard Reciprocal Recommendation baselines (beyond heuristic Uniform/MaxMatch) and optional adaptations of general long-term engagement / retention-oriented methods remains a gap.
>
> We would like to address a critical misunderstanding by the reviewer.
>
> **The lack of standard Reciprocal Recommendation baselines is strictly not a gap because we grant match probabilities as given.** Standard Reciprocal Recommendations calculate the match probabilities, then rank them in order of relevance. This is the exact same as Max Match, and thus **Max Match covers all works on standard Reciprocal Recommendations.**
>
> As we also mentioned in the above comment, we are not aware of any applicable “general long-term engagement / retention-oriented methods” (rather than fairness algorithms which are seemingly-possible solutions). Unless the reviewer can specify a concrete baseline and explain how it could provide additional value in our setting, it is difficult for us to assess its relevance or conduct a meaningful empirical comparison. We believe that the reviewer have such a method in mind when making the comment, and we would very much appreciate learning more about it. Without a clearly defined suggestion that can be integrated into our setup, however, we would have no option but to conclude that there is no identifiable gap in our current experimental design.
>
> > Note: The citation for FairCo is indeed missing in the experiment section (Page 7, line 347) .
>
> We appreciate the remark and will add the citation. FairCo is already introduced and cited in Section 2.2, so the missing citation in the experiment section does not affect attribution, but we agree adding it improves clarity.
>
> > Given that the rebuttal revision allows for 10 pages, I strongly suggest moving the Related Work section back to the main body.
>
> We appreciate the reviewer’s suggestion. We will move the Related Work section back to the main body in the revision. We initially placed it in the appendix due to space limitations, but we agree that including it in the main text will improve readability.

---

> ### Comment · Reviewer_ioZz · 2025-11-28
> **Response to Authors**
>
> I thank the authors for the detailed rebuttal and the updated draft.
>
> **Baselines & Storytelling**: While I maintain that static policies (e.g., Tomita & Yokoyama, 2024) are not strictly inapplicable to online settings due to the feasibility of batch approximations, I acknowledge the time constraints of the rebuttal period. In light of the additional experimental results provided (including the experiments in response to other reviewers, like the perturbation one), I tend to view the empirical evaluation as sufficient and no longer view the baseline selection as a major weakness. Regarding the storytelling, I still find that the Introduction places too much emphasis on the failure of fairness methods rather than the specific success mechanisms of MRet. However, I view this as a presentation choice rather than a fundamental flaw or rejection criterion.
>
> **Interpretation of FairCo (Remaining Disagreement):** I respectfully maintain my reservation regarding the interpretation of FairCo’s behavior in the synthetic setup. Fundamentally, FairCo is a *fairness-of-exposure* method designed to address position bias by ensuring exposure aligns with merit. As stated in its abstract, it *"explicitly enforces merit-based fairness guarantees"* by controlling exposure relative to a provided merit signal. Additionally, its problem definition (Sec. 4.1) formalizes an exposure-based fairness setting, rather than removing bias that is hard-coded into the underlying relevance/merit definition. In short,  FairCo is designed primarily to address **position bias** (i.e., handling the visibility weights $\alpha_k$) to ensure exposure is proportional to merit. The paper itself effectively models this and includes ablation studies on the examination function (e.g., variations in $\alpha_k$).
>
> In the synthetic experiments, popularity skew is explicitly injected into the ground-truth relevance $r(x,y)$ via the $\kappa$-dependent term. Since FairCo is designed to satisfy this **$\text{Exposure} \propto \text{Merit}$, and the provided "merit" ($r$) inherently encodes popularity skew, the algorithm is mathematically bound to allocate higher exposure to popular users.** This behavior reflects the algorithm faithfully optimizing its objective, rather than an algorithmic failure to address bias. Put differently, under this construction, the fairness objective is itself misaligned with mitigating popularity bias (in the common sense of exposure exceeding what underlying merit would justify).
>
> That said, I appreciate the authors’ thorough rebuttal and the additional experiments. I believe the proposed MRet method remains novel and technically sound, and I hope these comments help further clarify the interpretation and framing in the final version.

---

> ### Author Response · Authors · 2025-12-03
> **Official Comment by Authors**
>
> **We would like to thank the reviewer for confirming that all concerns other than those related to FairCo, including the baseline and storytelling aspects, have been fully resolved.**
>
> > Interpretation of FairCo
>
> It seems that the reviewer’s remaining concern is solely the issue regarding the handling of FairCo, and there still appears to be some misunderstanding by the reviewer. We discuss this in more detail below.
>
>
> **FairCo is a method that intends to solve popularity bias and position bias separately**. Specifically, popularity bias is resolved by setting and optimizing the exposure-merit fairness objective, and position bias is addressed specifically in the estimation of relevance using a regression estimator (Section 6.2, Morik et al. 2020) and importance sampling (Section 6.3, Morik et al. 2020). Therefore, we respectfully address the reviewer’s misunderstanding, as **“exposure aligns with merit” is not a solution for position bias but for popularity bias.**
>
>
> Now, the reviewer seems to be doubting the FairCo baseline, because popular users get more exposure (i.e., exposure fairness). **Therefore, we assume, from the arguments by the reviewer, that the reviewer is suggesting to compare with a baseline where all users have equal exposures. As an equal exposure baseline, we would like to note that the _Uniform_ baseline already represents the case where all users have equal exposure.**
>
> **In case, we further compare the performance with FairCo but the objective function targets equal exposure**. This would allow equal exposure while maximizing the number of matches. This can be done simply by taking away the “merit” component from the FairCo objective function. Now, the disparity function for FairCo (equal exposure) is described
>
> $$D_{\tau} (y_i, y_j)= \frac{1}{\tau}\sum_{t=1}^{\tau} Exp_t(y_i)-\frac{1}{\tau}\sum_{t=1}^{\tau}Exp_t(y_j)$$
>
> **This disparity formulation would result in a baseline where it aims to allocate equal exposure to all users, and no popular users getting more exposure.** The results below confirm that, while FairCo (equal exposure) performs slightly better than FairCo, the conclusions and discussions of our paper would remain the same. We would like to thank the reviewer for the discussion and we hope this addresses the concern. We will add this in the revision.
>
> **Table: User retention rate with comparison to FairCo**
> | method            | t = 500 | t = 1000 | t = 1500 | t = 2000 |
> | ----------------- | ------- | -------- | -------- | -------- |
> | Uniform           | 0.709   | 0.611    | 0.571    | 0.552    |
> | Max Match             | 0.715   | 0.613    | 0.570    | 0.545    |
> | Fairco | 0.724   | 0.636    | 0.602    | 0.585    |
> | FairCo (equal exposure)| 0.724   | 0.641    | 0.609    | 0.594    |
> | **MRet**          |**0.729**   | **0.657**    | **0.633**    | **0.624**    |
> | **MRet (best)**           | **0.742**   | **0.673**    | **0.655**    | **0.651**    |

---

### Official Review · Reviewer_3G7d · 2025-10-31

**Soundness:** 3
**Presentation:** 3
**Contribution:** 3
**Rating:** 6
**Confidence:** 3

**Summary:**

This manuscript proposes a novel problem setting focused on maximizing user retention in two-sided matching platforms, along with a dynamic learning-to-rank algorithm (MRet) that models personalized retention curves to optimize recommendations. The work addresses a critical gap in existing research—where match maximization causes user imbalance and fairness objectives fail to directly align with platform sustainability—and validates the approach through rigorous synthetic and real-world experiments on a large-scale online dating platform. The paper is well-structured, logically coherent, and its core contribution directly responds to practical needs of two-sided platforms dependent on user retention.

**Strengths:**

1. The paper breaks away from the dominant paradigms of match maximization and axiomatic fairness, instead centering on user retention as the core objective—an issue directly tied to platform revenue and sustainability (e.g., subscription-based models). This choice is well-justified by empirical evidence (Figure 1) showing that users with fewer matches have significantly higher churn rates, addressing a real pain point ignored by prior work.
2. MRet’s focus on dual retention gains (both the recommending and recommended users) is a key innovation. Unlike conventional methods that optimize for one-sided utility, this two-sided consideration aligns with the reciprocal nature of matching platforms. The use of personalized retention curves, derived from user profiles and interaction history, ensures recommendations are tailored to individual retention needs rather than generic fairness or match probability.
3. Experiments not only demonstrate higher retention rates but also explore edge cases (e.g., varying popularity skew, sparse match data, different exposure probabilities), proving MRet’s robustness. The analysis of match distribution (Figure 5) further clarifies why fairness-based methods fail—they ignore individual satisfactory match counts—strengthening the paper’s core argument.

**Weaknesses:**

1. The paper assumes match probabilities (r(x,y)) are either known or estimated upstream, but provides no guidance on how to integrate this estimation into the MRet framework. In practice, inaccurate match probability estimates could degrade retention performance, and the authors should discuss how MRet interacts with common estimation methods (e.g., collaborative filtering) or mitigate estimation errors.
2. Although the paper mentions potential applications in recruitment, experiments are limited to online dating. Job matching has distinct characteristics (e.g., asymmetric priorities between employers and job seekers, longer decision cycles) that may affect retention dynamics. Validating MRet on a recruitment dataset would strengthen its generalizability.
3. The paper focuses on retention rates as the key metric but does not address whether MRet impacts user satisfaction beyond continued login. For example, do users receiving "retention-optimized" matches report higher engagement quality (e.g., longer conversations, successful relationships in dating)? Supplementing with qualitative or secondary engagement metrics would enrich the evaluation.

**Questions:**

Note: Here are some questions I have after reading the paper. Since I am not very familiar with this field, the authors may choose to answer selectively, focusing on issues related to the paper’s contributions. I will also take into account comments from other reviewers when forming my final evaluation.

1. The retention function f(x,m) is assumed to be concave, but the paper notes MRet performs well on non-concave functions in real-world data. What specific properties of non-concave retention functions does MRet leverage to maintain effectiveness, and are there scenarios where non-concavity would significantly degrade performance?
2. How does MRet handle users with limited interaction history (e.g., new users) who lack sufficient data to train personalized retention curves? Is a fallback strategy (e.g., cluster-based retention curves) used, and how does this affect overall retention?
3. The paper mentions platform business models may prioritize one user side (e.g., paying male users in dating). How would MRet be modified to incorporate weighted retention objectives (e.g., 60% weight on male retention, 40% on female), and what trade-offs would arise from such prioritization?
4. In the real-world experiment, missing match probabilities are imputed using ALS. How sensitive is MRet’s performance to imputation errors? Would alternative imputation methods (e.g., matrix factorization with side information) yield better retention results?

---

> ### Author Response · Authors · 2025-11-19
> **Official Comment by Authors**
>
> We appreciate the valuable and thoughtful feedback from the reviewer. We respond to the concrete questions and comments in detail below.
>
> > The paper assumes match probabilities (r(x,y)) are either known or estimated upstream, but provides no guidance on how to integrate this estimation into the MRet framework. In practice, inaccurate match probability estimates could degrade retention performance, and the authors should discuss how MRet interacts with common estimation methods (e.g., collaborative filtering) or mitigate estimation errors.
>
> > In the real-world experiment, missing match probabilities are imputed using ALS. How sensitive is MRet’s performance to imputation errors?
>
> We appreciate the reviewer’s interesting remark and suggestion.
>
> As the reviewer points out, MRet may rely on the quality of the match probability estimation. **However, our framework is designed to be agnostic to the specific estimation method.** Our primary contribution is the novel formulation that directly optimizes user retention, a perspective lacking in prior works, rather than the estimation of match probabilities itself. We consider the improvement of match probability prediction an **orthogonal research topic**, and MRet can be seamlessly integrated with any estimator. **Furthermore, since the baselines optimize towards different objectives, it would be highly unlikely that MRet would lose its relative advantage over them even when match probability estimates are imperfect.**
>
> To empirically validate this robustness, we conducted an additional experiment where we injected noise into the match probabilities. Specifically, we apply noise to the probabilities by
> $$\tilde r(x,y) = r(x,y) + \varepsilon_{x,y}, \qquad \varepsilon_{x,y} \sim \mathcal{U}[-\delta,\delta],$$
> where ($\delta$) controls the noise magnitude. The results summarized in the table below show that MRet inherits the best performance to different estimation noises of the match probability. We will add the results in the revised version.
>
> **Table: User retention rate on different estimation noise**
> | method / $\delta$     | 0.0     | 0.1     | 0.2     | 0.3     | 0.4     | 0.5     |
> |-----------------|---------|---------|---------|---------|---------|---------|
> | Max Match       | 0.545   | 0.549   | 0.553   | 0.556   | 0.555   | 0.554   |
> | FairCo          | 0.585   | 0.582   | 0.581   | 0.577   | 0.578   | 0.581   |
> | Uniform         | 0.552   | 0.552   | 0.552   | 0.552   | 0.552   | 0.552   |
> | **MRet**        | **0.618** | **0.615** | **0.609** | **0.608** | **0.605** | **0.601** |
> | **MRet (best)** | **0.651**   | **0.646**   | **0.639**   | **0.633**   | **0.628**   | **0.625**   |
>
> > Would alternative imputation methods (e.g., matrix factorization with side information) yield better retention results?
>
> This is not confirmable in the real-data experiments, since ALS is used to define r(x, y) for those that are not filled. However, in the synthetic experiments where we know the true rewards, we confirmed that the better estimation of $r(x, y)$ results in better retention, as shown in the table above. Yet, MRet shows robustness to inaccurate estimations too.
>
> > The paper focuses on retention rates as the key metric but does not address whether MRet impacts user satisfaction beyond continued login. For example, do users receiving "retention-optimized" matches report higher engagement quality (e.g., longer conversations, successful relationships in dating)? Supplementing with qualitative or secondary engagement metrics would enrich the evaluation.
>
> This is a great question, and we are happy to clarify.
>
> We believe the reviewer has misunderstood our point that **we do not use user retention as a surrogate for user satisfaction**. We set user retention as the ultimate objective because platform sustainability relies heavily on it. We do not objectify engagement or user satisfaction. Therefore, addressing MRet impacts user satisfaction beyond retention would have limited value.
>
> However, even if the platform’s priority is to maximize user satisfaction or engagement, this can be done easily by replacing the reward signals from user retention with user satisfaction/engagement (e.g., explicit user satisfaction feedback, longer conversations, successful relationships in dating).

---

> ### Author Response · Authors · 2025-11-19
> **Official Comment by Authors (cont'd)**
>
> > asymmetric priorities between employers and job seekers
>
> > The paper mentions platform business models may prioritize one user side (e.g., paying male users in dating, longer decision cycles). How would MRet be modified to incorporate weighted retention objectives (e.g., 60% weight on male retention, 40% on female), and what trade-offs would arise from such prioritization?
>
> We would like to address the two insightful remarks jointly.
>
> In terms of asymmetric priorities, MRet can handle this simply by introducing a weight $w$ to the objective function
> $$\textstyle w(x)\big[f(x,m_{1:\tau}(x)+\sum_{k}\alpha_k r(x,\sigma_{\tau,k}))-f(x,m_{1:\tau}(x))\big] +\sum_{k} w(\sigma_{\tau,k})\big[f(\sigma_{\tau,k},m_{1:\tau}(\sigma_{\tau,k})+\alpha_k r(\sigma_{\tau,k},x))-f(\sigma_{\tau,k},m_{1:\tau}(\sigma_{\tau,k}))\big].$$
> Then, under the same concavity assumption as in Sec. 3.1, the score is argsorted by
> $$\text{Score}(y)=\frac{w(x)}{A}f\big(x,m_{1:\tau}(x)+Ar(x,y)\big) +\frac{w(y)}{\alpha_{\max}}\big[f\big(y,m_{1:\tau}(y)+\alpha_{\max} r(x,y)\big)-f\big(y,m_{1:\tau}(y)\big)\big].$$
> To analyze the trade-offs arising from such prioritization, we conducted an experiment using the weighted MRet formulation. We tested three weight scenarios for user groups X and Y: balanced ($w(x)=0.5, w(y)=0.5$), slightly biased ($0.6, 0.4$), and extremely biased ($0.9, 0.1$).
>
> **Table: User retention rates of each group under asymmetric weights**
> | Method            | w(x)=0.5, w(y)=0.5 | w(x)=0.6, w(y)=0.4 | w(x)=0.9, w(y)=0.1 |
> |------------------|--------------------|--------------------|--------------------|
> |                  | X / Y              | X / Y              | X / Y              |
> | Uniform          | 0.5555 / 0.5481    | 0.5555 / 0.5481    | 0.5555 / 0.5481    |
> | Max Match        | 0.5493 / 0.5410    | 0.5493 / 0.5410    | 0.5493 / 0.5410    |
> | FairCo (λ=100)   | 0.5849 / 0.5842    | 0.5849 / 0.5842    | 0.5849 / 0.5842    |
> | **MRet**         | **0.6243 / 0.6240** | **0.6198 / 0.6221** | **0.6233 / 0.6227** |
> | **MRet (best)**  | **0.6502 / 0.6515** | **0.6537 / 0.6507** | **0.6530 / 0.6498** |
>
> **Interestingly, the retention rates for both groups remained stable, even when we heavily favored one side ($w_x=0.9, w_y=0.1$). This result highlights a simple but key fact in two-sided markets: the two groups strongly depend on each other.** To keep users in Group X, the platform must also keep users in Group Y to provide enough matches. This suggests that MRet correctly captures this relationship. Instead of ignoring the lower-weighted group, the algorithm finds that keeping a balance is the best strategy to maximize the weighted objective. We will add this discussion and the formulation in the revision.
>
> Moreover, to answer the example raised by the reviewer about “longer decision cycles,” this depends totally on where the “match” is placed. If the “match” is placed when the applicant receives the job, this would be a much more difficult problem that should rather be solved as an independent study. However, the objective can be placed at when the two sides like each other in a job application. Then, delay would be no different from online dating.
>
> > The retention function f(x,m) is assumed to be concave, but the paper notes MRet performs well on non-concave functions in real-world data. What specific properties of non-concave retention functions does MRet leverage to maintain effectiveness, and are there scenarios where non-concavity would significantly degrade performance?
>
> This is a great question, and we are happy to clarify.
>
> As the reviewer points out, MRet retains the most users even when the function is not perfectly concave, as tested in the real-world experiments. While concavity is required for our theoretical guarantees, we could assume that typical real-world shapes are concave. Figure 1 shows that both male and female retention in our dataset follow this pattern. In such cases, the marginal-gain ordering given by MRet remains meaningful, even without exact concavity.
>
> A "strongly non-concave" situation implies highly oscillatory behavior. For example, a strongly non-concave behavior would be users having low retention probability after two matches, sharply higher retention after three, and then dropping low again after four. **However, we find it difficult to envision such strongly non-concave behavior in any realistic two-sided matching market. If the reviewer has a detailed example of a meaningfully general scenario where such fluctuations occur, we would need to understand them to have a more constructive discussion.** In the absence of such counterexamples, there is no option but to assume that our formulation and experiment cover the practical cases.

---

> > ### Author Response · Authors · 2025-11-19
> > **Official Comment by Authors (cont'd)**
> >
> > > How does MRet handle users with limited interaction history (e.g., new users) who lack sufficient data to train personalized retention curves? Is a fallback strategy (e.g., cluster-based retention curves) used, and how does this affect overall retention?
> >
> > We are happy to clarify. We use a **profile-based estimation** strategy that naturally covers new users without needing a complex fallback mechanism. As detailed in Appendix F, our experiments estimate retention curves using user clusters derived from static profile features (excluding interaction history). Since this method relies solely on user attributes available at registration, it functions effectively for both new and existing users. While our current experiments utilize this robust profile-based approach, the **MRet framework is agnostic to the underlying regression model**. In a production environment, one could straightforwardly extend the regressor to include interaction-based features (e.g., past match rates) as they become available, progressively improving accuracy for active users.

---

> > > ### Comment · Reviewer_3G7d · 2025-11-24
> > >
> > > Thanks to the authors for the very detailed and clear rebuttal. I’ve read through the responses, and they address my earlier points well. I especially appreciate the additional experiments on noisy match probabilities, the weighted retention analysis, and the clarification on handling new users. I hope the authors can incorporate these clarifications and additional results into the final version. Overall, the rebuttal strengthens the paper, and I am increasing my score by +2 (from 6 → 8).

---

### Author Response · Authors · 2025-12-04
**Summary of Revisions by Authors**

Dear Area Chair and reviewers.

We thank all reviewers for their constructive feedback. We are encouraged that:
* Reviewer 3G7d found that the paper “addresses a critical gap in existing research,” and found it “well‑structured” and “logically coherent.” **The reviewer explicitly raised the score (6→8) before the data leakage incident.**
* Reviewer M3j6 highlighted that our problem setting and formulation are “novel,” “clean, and actionable.”
* Reviewer WdJu described our model as “novel and with practical relevance,” noting that the theoretical relaxation is sound and that the empirical analysis is “very comprehensive.”
* Reviewer ioZz emphasized that MRet is “interesting” and “technically sound.”

We are particularly encouraged by the productive discussions during the rebuttal phase, that led Reviewer 3G7d to raise the score (6 → 8). **The reviewer explicitly stated in the official comment before the data leakage insident: “Overall, the rebuttal strengthens the paper, and I am increasing my score by +2 (from 6 → 8).” Reviewer **ioZz** has also confirmed that all their concerns are resolved positively.** There remains only a small misunderstanding of the baseline by the reviewer, which we addressed this along with additional experiments. While we also believe we have flawlessly addressed all the concerns by Reviewer **M3j6** and Reviewer **WdJu**, we were not able to receive any responses from them.

**Overall, the main concerns by all reviewers are now resolved in the revised manuscript. We appreciate the meta-review process and the consideration given these increasingly positive assessments.**

---

Below we also summarize the key updates, which are marked in red in the revised manuscript:

* We empirically demonstrated that **MRet remains the best solution even when there is noise $\delta$ in the match probabilities.**

* We added experimental results with a **larger sample size 5000 x 5000** in the appendix.

* We moved the Related Works section to the main text, with some more references to the related literature.

* In the experiments section (Section 4.1), we added an explanation of why [Tomita and Yokoyama, 2024] cannot be compared as a baseline.

* We added experimental results with weighted retention objectives $w(x)$ and $w(y)$ in the appendix.

* We added experimental results with a uniform exposure version of FairCo in the appendix.

* Reworded from “popularity bias” to “popularity skew”

* Added citation to FairCo in the experiments section (Section 4.1).

**We believe these updates fully resolve the reviewers’ main concerns and significantly strengthen the manuscript. We thank the reviewers again for the thoughtful and valuable feedback.**

---

### Meta-Review · Area_Chair_wNQG · 2026-01-07

**Summary:**

The paper proposes a way of maximizing user retention in two-side matching platforms, as an alternative of maximizing the total number of matches. The authors develop a new objective that measures user retention and propose a method for optimizing it. Experiments on synthetic and real data show that the proposed method performs better in user retention than methods based on match maximization and fairness-enforcement.

The paper received mixed reviews. Overall, the reviewers appreciated the studied problem and highlighted the clarity of exposition of the technical details.

At the same time, several reviewers raised concerns regarding the comparison against fairness enducing methods. They also highlighted several strong modeling assumptions (e.g. known relevances, assumptions needed to compute the retention function etc). Finally, reviewers M3j6 and ioZz also argued that for the particular fairness measure studied by the baseline method, the fairness objective itself may be in conflict with user retention (as it explicitly seeks to allocate exposure in proportion to merit).

The authors do acknowledge in their paper that their method outperforming the baselines is not surprising, as these don't optimize for user retention explicitly. Overall, the discussions positioned the paper more clearly with respect to prior work, with reviewer ioZz accepting the baselines as reasonable. The authors are encouraged to incorporate the reviewers' feedback and the discussions in the next version of the paper. In particular, they should elaborate on comparisons with prior work (possibly on one-sided user-item recommender systems) on user retention and other forms of user engagement maximization and possibly also works on enforcing diversity of recommendations.

**Reviewer Concerns:**

The authors did address several concerns regarding ablation studies and larger experiments. On the other hand, the justification regarding the NP does not seem sufficiently formal and should be strengthened in the next version of the paper. The justifications regarding the comparison with fairness-inducing methods and the choice of fairness metric do go towards clarifying the scope of the current experiments. However, positioning more clearly with respect to prior work on user retention, other forms of user engagement maximization and diversity of recommendations seems necessary.

**Reviewer Scores:**

Reviewer 3G7d indicated they would change their score to 8. Reviewer ioZz engaged in discussions, but didn't raise. Therefore, it is unlikely they would have changed scores substantially.

The other two reviewers did not respond. Reviewer M3j6 expressed concerns about the fairness metric potentially harming retention, similarly to Reviewer ioZz, who did not raise. Therefore, I think it is unlikely Reviewer M3j6 would have raised. Reviewer WdJu mostly focused on requesting additional experiments and discussions and gave a low confidence of 2. Therefore, they might have raised to 6, but I do not expect they would have strongly supported acceptance.

---

### Decision · Program_Chairs · 2026-01-26

Accept (Poster)